# Probabilistic Robustness Certificates against Adversarial Attacks

**Sara Taheri** [1]   **Majid Zamani** [1][2]

## Abstract

The growing use of machine learning in safety-critical settings increases vulnerability to *adversarial attacks*. Existing defense mechanisms typically either lack formal guarantees or depend on restrictive assumptions about the model family, the threat model, or the poisoning budget, and many only offer point-wise certification. Importantly, they often overlook the inherent stochasticity of modern training pipelines, which undermines their practical reliability. In this work, we introduce a probabilistic framework that views gradient-based training as a *discrete-time stochastic dynamical system* and formulates poisoning robustness as a safety verification task. Using *barrier certificates* (BC), we derive sufficient conditions to probabilistically certify a robust radius against worst-case $\ell_p$-bounded poisoning, guaranteeing that the final model parameters remain within a safe set probabilistically. For tractable computation, we represent BCs with neural networks and obtain *probably approximately correct* (PAC) guarantees through a *scenario convex problem*. Our approach determines the maximum certified radius within which the trained model achieves probabilistic accuracy at a pre-specified confidence level. Experiments on MNIST, SVHN, and CIFAR-10 show that our framework offers robustness guarantees under stochastic training, while being model-agnostic and not requiring knowledge of the attack strategy.

## 1. Introduction

As machine learning (ML) systems are increasingly deployed in safety-critical domains such as autonomous driving, medical diagnostics, and industrial control, their vulnerability to adversarial manipulation has become a pressing concern. These applications demand not only high predictive performance, but also formal guarantees that the system will behave reliably under adversarial conditions. Among the most critical threats are **data poisoning attacks**, in which an adversary subtly alters a subset of the training data to induce test-time failures or degrade model performance. Such attacks compromise the training process itself, embedding stealth backdoors or vulnerabilities that can remain latent until triggered at deployment (Carlini et al., 2024; Schwarzschild et al., 2021). This difficulty is compounded by the intrinsic randomness of ML pipelines and the frequently opaque nature of their training processes. Multiple sources contribute to this randomness, including optimizer-level noise, stochastic data augmentation, architectural randomness, and external factors such as random initialization. These factors collectively induce stochastic training trajectories, leading to models whose final parameters may vary between runs. Although numerous defenses have been proposed, including poisoned-sample detection, robust loss functions, and adversarial training, most are *heuristic* and break down under adaptive strategies (Huang et al., 2020; Koh et al., 2022; Goldblum et al., 2023). These challenges motivate the need for **formal certification frameworks** that can provide provable *probabilistic* guarantees of robustness against bounded poisoning at both training and test time, when the underlying training process is stochastic.

**Related Work.** Most robustness certification efforts focus on *test-time evasion*; in contrast, formal guarantees for *train-time poisoning* are less developed, and explicit treatment of *stochastic* training—i.e., randomness induced by the end-to-end pipeline rather than a single deterministic run—remains uncommon. To the best of our knowledge, there is no unified framework that certifies robustness against *both* train-time poisoning and test-time perturbations while modeling training itself as a stochastic process. Previous work, typically developed under deterministic or otherwise restricted regimes, can be grouped as follows. **(i) Ensemble-based certifications** aggregate predictions from independently trained models (e.g., through partitioning/subsampling) and derive probabilistic guarantees through voting (Levine & Feizi, 2021; Jia et al., 2021; Rezaei et al., 2023); such guarantees commonly hinge on structural assumptions about data splitting and the adversary, and often operate by aggregating *pointwise* certificates. **(ii) Randomized smoothing**

---
[1]Department of Computer Science, LMU Munich, Germany [2]Department of Computer Science, CU Boulder, USA. Correspondence to: Sara Taheri <sara.taheri@lmu.de>, Majid Zamani <majid.zamani@colorado.edu>.

*Proceedings of the $43^{rd}$ International Conference on Machine Learning*, Seoul, South Korea. PMLR 306, 2026. Copyright 2026 by the author(s).

**for poisoning/backdoors** extends smoothing-style ideas by injecting additional randomness into learning or prediction (Weber et al., 2023; Wang et al., 2020; Rosenfeld et al., 2020; Zhang et al., 2022), yielding probabilistic (often per-instance) certificates under prescribed noise/attack models, but typically without certifying the *training procedure* as a stochastic dynamical system that captures training-time randomness. **(iii) Privacy-inspired robustness** leverages connections between differential privacy and robustness (Ma et al., 2019; Xie et al., 2023); however, the resulting robustness is often implicit, may incur accuracy trade-offs, and does not directly provide explicit $\ell_p$-budget certified radii. **(iv) Model- and architecture-specific certificates** target particular hypothesis classes or corruption models, including kernel/NTK-based analyses (Gosch et al., 2025; Sabanayagam et al., 2025), convex/abstract relaxations (Sosnin et al., 2025), and certificates specialized to decision trees (Meyer et al.), nearest neighbors (Jia et al., 2022), or GNNs (Sabanayagam et al., 2025; Gosch et al., 2025). Although strong within scope, these approaches often rely on restrictive assumptions (e.g., idealized or deterministic training, white-box access, or precise knowledge of threat models/budgets). Across these lines, recurring limitations include the reliance on specific threat-model or budget constraints (often assumed known) (Weber et al., 2023), model/access assumptions (Meyer et al.; Jia et al., 2022; Sabanayagam et al., 2025; Gosch et al., 2025), predominantly pointwise rather than distribution-level guarantees (Levine & Feizi, 2021), and limited coverage of optimization-driven train-time poisoning under stochastic training pipelines. These gaps motivate the question:

> *"Is it possible to provide, with probability at least $\rho$, a formal certification of an $\ell_p$-bounded poisoning budget such that, despite stochastic training dynamics, the performance of the resulting model is guaranteed to stay above a given threshold $\alpha$?"*

**Proposed Framework.** We answer this question by developing a formal robustness certification framework based on *control-theoretic safety verification* (Taheri et al., 2026). We cast gradient-based training as a *discrete-time stochastic dynamical system* (dt-SDS) whose parameter evolution is driven by adversarially perturbed data and intrinsic pipeline randomness. In this view, poisoning robustness becomes a probabilistic safety question, and the notions of **barrier certificates (BC)** (Prajna et al., 2007; Ames et al., 2014) are extended to certify an admissible poisoning radius by separating safe/unsafe regions in the parameter space. To address the intractability of high-dimensional training dynamics, the certificate is parameterized as a neural network (NNBC) and learned from sampled realizations of stochastic training trajectories (Anand & Zamani, 2023), avoiding any explicit model of the underlying dynamics. To improve scalability, we further allow the NNBC synthesis stage to use

a lightweight surrogate transition model trained from limited rollouts of the true stochastic training dynamics. This surrogate is used only to generate additional synthesis trajectories and is not used for the final certification step, thereby separating computationally efficient certificate construction from formal validation. To certify generalization beyond the training samples, we verify the learned NNBC via a *scenario convex problem* (SCP), a tractable relaxation of the *robust convex problem* (RCP). This step yields *probably approximately correct* (PAC) guarantees (Campi & Garatti, 2008) on the validity of the robustness certificate. In particular, we certify the largest admissible $\ell_p$-norm perturbation budget such that, at a prescribed confidence level, the ML model remains accurate under the perturbation within this budget, up to a violation probability.

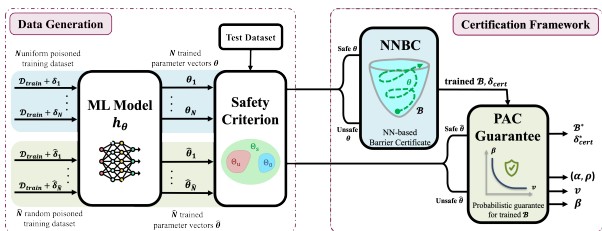

*Figure 1.* Proposed framework: we construct an NNBC $\mathcal{B}$ from stochastic training trajectories. By applying PAC-style generalization bounds, we guarantee that, with confidence $1 - \beta$, the model $h_x$ is $(\alpha, \rho)$-certified accurate within a derived robust radius.

**Key Contributions. 1.** We model gradient-based ML training as a discrete-time *stochastic dynamical system*. Then, robustness certification against train- and test-time attack is reformulated as a *probabilistic safety verification* problem using *barrier certificates (BCs)*. **2.** We introduce a neural network-based BC (NNBC) framework to overcome the intractability of explicit BC design for high-dimensional and unknown poisoned training dynamics. NNBC is trained to obtain the certified robust radius, the largest admissible perturbation of the training or test data for which the ML model is certified accurate. **3.** We develop a surrogate-assisted synthesis strategy that improves scalability by using a lightweight transition model to generate additional trajectories for candidate NNBC construction, while retaining formal certification through verification on fresh samples from the true stochastic dynamics. **4.** We derive a *probably approximately correct* (PAC) bound that provides a rigorous probabilistic guarantee for the trained NNBC and its associated certified robust radius. **5.** Our method is fully *model-* and *attack-agnostic* and requires no prior knowledge of the model architecture, poisoning strategy, or corruption level, ensuring broad applicability across stochastic ML pipelines. **6.** We empirically validate our framework on multiple models and datasets, demonstrating that it can formally certify non-trivial poisoning radii within which performance remains robust under stochastic training, even in the presence of fully-unknown training dynamics.

## 2. Preliminaries

All proofs and notation are deferred to Appendix B.

### 2.1. Setup Formulation

We consider a supervised learning problem defined on a clean training dataset $\mathcal{D}_{\mathrm{tr}} = \{(u_i, y_i)\}_{i=1}^n \subseteq \mathbb{R}^m \times \mathcal{Y}$, where $\mathcal{Y} = \{1, \ldots, k\}$ is the set of class labels and a held-out test dataset $\mathcal{D}_{\mathrm{te}} = \{(u_i', y_i')\}_{i=1}^{n'} \subseteq \mathbb{R}^m \times \mathcal{Y}$ for evaluation. Each feature vector $u_i \in \mathbb{R}^m$ is paired with a label $y_i \in \mathcal{Y}$. Let $h_x : \mathbb{R}^m \to \mathbb{R}^k$ be a parameterized ML model (e.g., a neural network) with parameters $x \in \mathbb{R}^d$, mapping inputs to a continuous score vector (e.g., logits). The model is trained by minimizing the empirical loss $\mathcal{L}(h_x, \mathcal{D}_{\mathrm{tr}}) := \frac{1}{n} \sum_{i=1}^n \ell(h_x(u_i), y_i)$, where $\ell : \mathbb{R}^k \times \mathcal{Y} \to \mathbb{R}_{\geq 0}$ is a nonnegative pointwise loss (e.g., cross-entropy). Training proceeds through iterative gradient-based updates, potentially under multiple sources of stochasticity. The output of $h_x$ is a continuous $k$-dimensional score vector rather than a label in $\mathcal{Y}$; the discrete prediction is obtained only when needed (e.g., at evaluation) by $\arg\max$. While standard training assumes clean data, an adversary may corrupt inputs at train-time or test-time to degrade performance or induce unsafe behavior. We focus on input-space attacks that perturb feature vectors while keeping labels unchanged, and formalize the threat model next.

**Definition 2.1** (Adversarial Attacks). Let $\mathcal{D}_{\mathrm{tr}} = \{(u_i, y_i)\}_{i=1}^n$ and $\mathcal{D}_{\mathrm{te}} = \{(u_i', y_i')\}_{i=1}^{n'}$ be the clean train and test datasets, respectively. A *train-time poisoning attack* is an adversary $\mathcal{A}$ that perturbs a fraction $\tau \in [0, 1]$ of training inputs ($r := \lceil \tau n \rceil$ samples), i.e.,

$$\mathcal{D}_{\mathrm{tr}}^{\Delta} := \{(u_i + \delta_i, y_i)\}_{i=1}^r \cup \{(u_i, y_i)\}_{i=r+1}^n, \quad (1)$$

where $\Delta := [\delta_1 \ldots \delta_r] \in \mathbb{R}^{r \times m}$ is the poisoning perturbation matrix, and $\Omega := \{\Delta \in \mathbb{R}^{r \times m} \mid \|\Delta\|_p := \max_{i \in [r]} \|\delta_i\|_p \leq \delta\}$ denotes the set of admissible train-time perturbations for some $\delta \geq 0$. Similarly, a *test-time evasion attack* is an adversary $\mathcal{A}'$ that perturbs a fraction $\tau' \in [0, 1]$ of test inputs ($r' := \lceil \tau' n' \rceil$ samples), i.e.,

$$\mathcal{D}_{\mathrm{te}}^{\Delta'} := \{(u_i' + \delta_i', y_i')\}_{i=1}^{r'} \cup \{(u_i', y_i')\}_{i=r'+1}^{n'}, \quad (2)$$

where $\Delta' := [\delta_1' \ldots \delta_{r'}'] \in \mathbb{R}^{r' \times m}$ is the evasion perturbation matrix, and $\Omega' := \{\Delta' \in \mathbb{R}^{r' \times m} \mid \|\Delta'\|_p := \max_{i \in [r']} \|\delta_i'\|_p \leq \delta'\}$ denotes the set of admissible test-time perturbations for some $\delta' \geq 0$. Without loss of generality, we assume that the first $r$ (resp. $r'$) elements of $\mathcal{D}_{\mathrm{tr}}$ (resp. $\mathcal{D}_{\mathrm{te}}$) are perturbed. In addition, if either $\delta = 0$ or $\tau = 0$ (resp. $\delta' = 0$ or $\tau' = 0$), then $\mathcal{D}_{\mathrm{tr}}^{\Delta} = \mathcal{D}_{\mathrm{tr}}$ (resp. $\mathcal{D}_{\mathrm{te}}^{\Delta'} = \mathcal{D}_{\mathrm{te}}$).

Therefore, we evaluate the generalization performance of the ML model $h_x$ on a test dataset $\mathcal{D}_{\mathrm{te}}^{\Delta'}$ via:

$$\mathcal{G}(x) := \frac{1}{n'} \sum_{i=1}^{n'} \mathbf{1}_{\left\{ \arg\max_{j \in [k]} \left( h_x(u_i' + \delta_i') \right)_j = y_i' \right\}}, \quad (3)$$

which returns the fraction of correctly classified test samples. Here, $\delta_i' \in \mathbb{R}^m$ is the perturbation applied to the $i$-th input in $\mathcal{D}_{\mathrm{te}}^{\Delta'}$, with $\delta_i' = \mathbf{0}$ for all unperturbed samples ($i > r'$) as in Definition 2.1. We assess the trained model by its final test accuracy $\mathcal{G}(x(T))$, where $x(T)$ denotes the parameters after training iterations $T$. Train-time poisoning can steer the learning trajectory toward suboptimal or unsafe parameter regions, while test-time evasion can induce misclassification at deployment, motivating robustness certification with formal guarantees under such perturbations.

**Definition 2.2** (Certified Accuracy). Let $h_x$ be an ML model trained on a (possibly) poisoned dataset $\mathcal{D}_{\mathrm{tr}}^{\Delta}$ and evaluated on the (possibly) perturbed test dataset $\mathcal{D}_{\mathrm{te}}^{\Delta'}$ using $\mathcal{G}$ in (3). Given constants $\alpha, \rho \in (0, 1]$, the trained ML model $h_x$ is said to be $(\alpha, \rho)$-certified accurate if, with a probability of at least $\rho$, its final test accuracy remains above the target threshold $\alpha$, i.e.,

$$\mathbb{P}\big[ \mathcal{G}(x(T)) \geq \alpha \big] \geq \rho, \quad (4)$$

where the probability is taken over the randomness inherent in the training procedure, which induces the distribution of the terminal parameter $x(T)$.

Using Definitions 2.1 and 2.2, we now formally state our main problem below and tackle it in the following section.

**Problem 2.3.** Let $h_x$ be an ML model trained on a (potentially) poisoned dataset $\mathcal{D}_{\mathrm{tr}}^{\Delta}$ and evaluated on a (potentially) perturbed test dataset $\mathcal{D}_{\mathrm{te}}^{\Delta'}$. Given constants $\alpha, \rho \in (0, 1]$, the goal is to formally determine the largest train-time radius $\delta_{\mathrm{cert}}$ (resp. test-time radius $\delta_{\mathrm{cert}}'$) such that for any perturbations $\|\Delta\|_p \leq \delta_{\mathrm{cert}}$ (resp. $\|\Delta'\|_p \leq \delta_{\mathrm{cert}}'$), the resulting model is $(\alpha, \rho)$-certified accurate.

### 2.2. Methodology

We present a formal certification framework for stochastic training by taking a systems-theoretic perspective that characterizes training as a stochastic dynamical system. This formulation accounts for stochasticity—stemming from mini-batch sampling, data augmentation, stochastic regularization, and random initialization—and supports a rigorous analysis of how randomness and adversarial perturbations together influence the model parameters. The next definition makes this representation precise.

**Definition 2.4** (dt-SDS). Let $h_x$ be an ML model as defined in Section 2.1. A discrete-time stochastic dynamical system (dt-SDS) corresponding to the ML model is a tuple $\mathfrak{S} = (X, X_0, \mathcal{D}_{\mathrm{tr}}^{\Delta}, \mathcal{W}, w, f)$, where $\mathcal{D}_{\mathrm{tr}}^{\Delta} \subseteq \mathbb{R}^m \times \mathcal{Y}$ is a (potentially poisoned) training dataset, $X \subseteq \mathbb{R}^d$ is the set of model parameters, $X_0 \subseteq X$ is the set of initial model parameters, $w$ denotes a sequence of independent and identically distributed (i.i.d.) random variables drawn from an uncertainty space $\mathcal{W}$, and $f : \mathbb{R}^d \times \mathbb{R}^m \times \mathcal{Y} \times \mathcal{W} \to \mathbb{R}^d$ is

the parameter update map and $\mathfrak{S}$ evolves according to:

$$x(t+1) = f\big(x(t), \mathcal{D}_{\mathrm{tr}}^{\Delta}(t), w(t)\big), \quad \forall t \in \mathbb{N}_0, \quad (5)$$

where $x(t) \in X$ and $\mathcal{D}_{\mathrm{tr}}^{\Delta}(t) \subseteq \mathcal{D}_{\mathrm{tr}}^{\Delta}$ denote the state of the system (model parameters) and the input of the system (mini-batch sampled), respectively, at iteration $t$. For any $x(0) \in X_0$, the induced training trajectory is $\xi = \langle x(0), x(1), x(2), \dots \rangle$.

For simplicity, we write the update map as $f(x, \Delta, w) := f\big(x, \mathcal{D}_{\mathrm{tr}}^{\Delta}, w\big)$, where $\Delta$ encodes the perturbation matrix applied to the training inputs. In this dynamical system view, robustness certification for $h_x$ can be cast as a *safety verification* problem: stochastic training induces parameter trajectories in $X$, and the model is deemed robust if, with a prescribed probability, these trajectories remain in the *safe* region of parameter space corresponding to satisfactory test accuracy. This reformulation enables the use of safety verification tools to derive rigorous robustness guarantees.

**Definition 2.5** (Certified Accuracy as a Safety Specification). Consider a dt-SDS $\mathfrak{S}$ describing the training process of an ML model $h_x$ as in Definition 2.4, with a test accuracy function $\mathcal{G}$ as in (3). Given constants $\alpha, \rho \in (0, 1]$, define the sets of safe and unsafe parameters with respect to $\alpha$ as

$$X_s^{\alpha} := \{ x \in X \mid \mathcal{G}(x) \geq \alpha \}, \quad X_u^{\alpha} := X \setminus X_s^{\alpha}. \quad (6)$$

The safety specification is indicated by the tuple $\Psi = (X_0, X_s^{\alpha}, X_u^{\alpha}, T, \rho)$, which characterizes trajectories starting from $X_0$ that remain in the safe set $X_s^{\alpha}$ at the final iteration $T$ with a probability of at least $\rho$. Satisfaction of $\Psi$ implies that the ML model $h_x$ achieves the test acuracy above $\alpha$ with a probability of at least $\rho$ as in Definition 2.2.

To formally verify the safety specification $\Psi$ for $\mathfrak{S}$ (equivalently, the certified accuracy), we utilize barrier certificates, following the approach of (Steinhardt & Tedrake, 2012), as the basis of our robustness certification framework.

**Definition 2.6** (BC). Let $h_x$ be an ML model as in Section 2.1, with associated dt-SDS $\mathfrak{S}$ as in Definition 2.4 and safety specification $\Psi$ as in Definition 2.5. A function $\mathcal{B} : \mathbb{R}^d \times \mathbb{N}_{0:T} \to \mathbb{R}_{\geq 0}$ is called a barrier certificate (BC) for $\mathfrak{S}$ with respect to the initial set $X_0 \subseteq X$ and the unsafe set $X_u^{\alpha} \subseteq X$, if there exist constants $b_1, b_2, b_3 \in \mathbb{R}_{>0}$, $\kappa \in (0, 1]$, and certified robust radii $\delta_{\mathrm{cert}}$ for train-time (resp. $\delta'_{\mathrm{cert}}$ for test-time) such that, for all perturbations with $\|\Delta\|_p \leq \delta_{\mathrm{cert}}$ (resp. $\|\Delta'\|_p \leq \delta'_{\mathrm{cert}}$), the following conditions hold:

$$\mathcal{B}(x, 0) \leq b_1, \qquad \forall x \in X_0, \quad (7)$$
$$\mathcal{B}(x, T) \geq b_2, \qquad \forall x \in X_u^{\alpha}, \quad (8)$$
$$\mathbb{E}\big[\mathcal{B}(x^+, t) \,|\, x\big] \leq \kappa \mathcal{B}(x, t-1) + b_3, \quad \forall x \in X, \quad (9)$$
$$b_1 \kappa^T + b_3 T \leq b_2 (1 - \rho), \quad (10)$$

for all $t \in \mathbb{N}_{1:T}$, where $x^+ = f(x, \Delta, w)$.

The time-dependent form of $\mathcal{B}$ allows the certificate to evolve throughout the training iterations and to certify *terminal* robustness, i.e., robustness of the final parameters rather than of the entire trajectory. The constants $b_1, b_2, b_3$ and $\kappa$ govern the initial bound, terminal separation, and expected contraction of the certificate, respectively.

*Remark* 2.7. While the BC conditions are the same for both train-time and test-time attacks, the underlying sources of variability are distinct. Train-time poisoning perturbs the dynamics $f(x, \Delta, w)$, thereby changing the distribution of parameter trajectories. By contrast, test-time evasion only alters $\mathcal{D}_{\mathrm{te}}^{\Delta'}$ and thus influences the safety evaluation exclusively through $\mathcal{G}(x)$. Other elements—such as the model architecture, choice of optimizer, attack intensity, and data distribution—also impact these trajectories. Crucially, because the BC constrains only the resulting parameter trajectories, it does not depend on where this variability comes from, which is a central strength of our framework.

We now present our main robustness guarantee for $h_x$.

**Theorem 2.8.** *Let $h_x$ be an ML model trained on a (potentially) poisoned dataset $\mathcal{D}_{\mathrm{tr}}^{\Delta}$ and evaluated on a (potentially) perturbed test dataset $\mathcal{D}_{\mathrm{te}}^{\Delta'}$. Given constants $\alpha, \rho \in (0, 1]$, consider the dt-SDS $\mathfrak{S} = (X, X_0, \mathcal{D}_{\mathrm{tr}}^{\Delta}, \mathcal{W}, w, f)$ as in Definition 2.4 and the safety specification $\Psi = (X_0, X_s^{\alpha}, X_u^{\alpha}, T, \rho)$ as in Definition 2.5. Suppose that there exists a BC $\mathcal{B}$ satisfying conditions (7)–(10) for all perturbations $\|\Delta\|_p \leq \delta_{\mathrm{cert}}$ (resp. $\|\Delta'\|_p \leq \delta'_{\mathrm{cert}}$) and for all $t \in \mathbb{N}_{1:T}$, with constants $b_1, b_2, b_3 \in \mathbb{R}_{>0}$ and $\kappa \in (0, 1]$. Then $h_x$ is $(\alpha, \rho)$-certified accurate as in Definition 2.2.*

Constructing a valid BC $\mathcal{B}$ is in general computationally intractable: the parameter space is high-dimensional, the dynamics $f$ do not admit a closed-form description, and the associated noise distribution is unknown. To overcome these challenges, we employ a data-driven synthesis approach that infers $\mathcal{B}$ from the training trajectories and pairs it with a formal validation step that certifies the resulting certificate.

## 3. Data-Driven Robustness Certification

We now develop a data-driven certification framework for $h_x$ by training a neural network barrier certificate (NNBC).

### 3.1. Data Generation

Our framework relies on two datasets: (i) a trajectory dataset to enforce the conditions in Definition 2.6, and (ii) an empirical mean dataset to approximate the expectation term in the constraint (9) using realizations of $w$.

**Trajectory Dataset.** To generate the trajectory dataset, we first define two sets of $N_b$ uniformly spaced poisoning budgets: $\delta_{\mathrm{grid}} = \langle \delta_1, \dots, \delta_{N_b} \rangle$ and $\delta'_{\mathrm{grid}} = \langle \delta'_1, \dots, \delta'_{N_b} \rangle$, where each point in $\delta_{\mathrm{grid}}$ and $\delta'_{\mathrm{grid}}$ represents train- and test-time poisoning levels, respectively. Depending on the certification type (train-time or test-time), one of the grids is

fixed at zero. For each index $i \in [N_b]$, we sample an initialization $x_i(0) \in X_0$. We then generate a specific poisoning perturbation $\Delta_i$ (resp. $\Delta_i'$) such that $\|\Delta_i\|_p = \delta_i$ (resp. $\|\Delta_i'\|_p = \delta_i'$). We train $h_x$ for $T$ iterations on the poisoned dataset $\mathcal{D}_{\text{train}}^{\Delta_i}$ to obtain the full trajectory of the parameters $\xi_i = \langle x_i(0), \ldots, x_i(T) \rangle$. The final trained model is then evaluated on the perturbed test dataset $\mathcal{D}_{\text{test}}^{\Delta_i'}$, which produces the test accuracy $\mathcal{G}(x_i(T))$. This procedure yields four subsets of state-time tuples used for training the NNBC:

$$\mathcal{I} = \big\{ x_i(0) \,\big|\, x_i(0) \in X_0 \big\}, \tag{11}$$

$$\mathcal{S} = \big\{ x_i(T) \,\big|\, \mathcal{G}(x_i(T)) \geq \alpha \big\}, \tag{12}$$

$$\mathcal{U} = \big\{ x_i(T) \,\big|\, \mathcal{G}(x_i(T)) < \alpha \big\}, \tag{13}$$

$$\mathcal{X} = \big\{ (x_i(t), t) \,\big|\, i \in [N_b], \ t \in \mathbb{N}_{1:T} \big\}, \tag{14}$$

where $\mathcal{I} \subseteq X_0$, $\mathcal{S} \subseteq X_s^\alpha$, and $\mathcal{U} \subseteq X_u^\alpha$. From this dataset, we compute the empirical robust radius $\delta_{\text{emp}}$ as the largest value in $\delta_{\text{grid}}$ (resp. $\delta_{\text{emp}}'$ as the largest in $\delta_{\text{grid}}'$) such that, for all perturbations $\Delta_i$ (resp. $\Delta_i'$) with $\|\Delta_i\|_p \leq \delta_{\text{emp}}$ (resp. $\|\Delta_i'\|_p \leq \delta_{\text{emp}}'$), the fraction of terminal states satisfying $\mathcal{G}(x_i(T)) \geq \alpha$ is at least $\rho$, i.e.,

$$\delta_{\text{emp}} := \max\Big\{ \delta \,\Big|\, \frac{\big|\{i \,|\, \delta_i \leq \delta, \mathcal{G}(x_i(T)) \geq \alpha\}\big|}{\big|\{i \,|\, \delta_i \leq \delta\}\big|} \geq \rho \Big\}, \tag{15}$$

$$\delta_{\text{emp}}' := \max\Big\{ \delta' \,\Big|\, \frac{\big|\{i \,|\, \delta_i' \leq \delta', \mathcal{G}(x_i(T)) \geq \alpha\}\big|}{\big|\{i \,|\, \delta_i' \leq \delta'\}\big|} \geq \rho \Big\}, \tag{16}$$

where $\delta \in \delta_{\text{grid}}$, $\delta' \in \delta_{\text{grid}}'$, and $i \in [N_b]$, with non-zero denominators. The quantities $\delta_{\text{emp}}$ and $\delta_{\text{emp}}'$ are empirical, sample-dependent estimates of the largest perturbation levels satisfying the $(\alpha, \rho)$ accuracy criterion over the generated trajectories. We use them only to initialize the candidate-radius search in the synthesis stage.

**Empirical Mean Dataset.** To approximate the expectation in (9), we take each state–time pair $(x_i, t) \in \mathcal{X}$ and generate $M$ i.i.d. next-state samples by drawing independent noise realizations $w_1, \ldots, w_M \sim \mathcal{W}$ and applying one update step via (5). We then replace the expectation with the empirical mean. This empirical substitution incurs an error. In the sequel, we use Chebyshev's inequality to control this error and obtain a rigorous relaxation of (9).

### 3.2. Neural Network BC (NNBC)

Now, we parameterize a BC with a neural network and train it to satisfy the conditions in Definition 2.6.

**Structure.** Given a dt-SDS $\mathfrak{S}$, we model the NNBC as $\mathcal{B}_\varphi : \mathbb{R}^d \times \mathbb{N}_{0:T} \to \mathbb{R}_{\geq 0}$. The network takes as input the state vector $x \in \mathbb{R}^d$ and the time $t \in \mathbb{N}_{0:T}$, and the output is a scalar barrier value. The set of training parameters $\varphi$ comprises the weights and biases of the network, as well as the BC constants ($b_1, b_2, b_3, \epsilon_M \in \mathbb{R}_{>0}$ and $\kappa \in (0, 1]$). We use ReLU activations in the hidden layers and an identity activation in the output layer. In addition, the depth and width are tunable hyperparameters.

**Loss Function.** To train an NNBC $\mathcal{B}_\varphi$ that satisfies conditions (7)–(10) for a candidate radius $\delta_{\text{cand}} \leq \delta_{\text{emp}}$ (resp. $\delta_{\text{cand}}' \leq \delta_{\text{emp}}'$), we minimize a composite loss over the datasets $\mathcal{I}, \mathcal{U}$, and $\mathcal{X}$ in (11)–(14), generated under perturbations with $\|\Delta_i\|_p \leq \delta_{\text{cand}}$ and $\|\Delta_i'\|_p \leq \delta_{\text{cand}}'$:

$$
\begin{aligned}
\mathcal{L} := {} & c_\rho \, \text{ReLU} \big( b_1 \kappa^T + b_3 T - b_2(1-\rho) \big) \\
& + c_n \sum_{(x_i,t) \in \mathcal{X}} \text{ReLU} \big( -\mathcal{B}_\varphi(x_i, t) \big) \\
& + c_\mathcal{I} \sum_{x_i \in \mathcal{I}} \text{ReLU} \big( \mathcal{B}_\varphi(x_i, 0) - b_1 \big) \\
& + c_\mathcal{U} \sum_{x_i \in \mathcal{U}} \text{ReLU} \big( b_2 - \mathcal{B}_\varphi(x_i, T) \big) \\
& + c_f \sum_{(x_i,t) \in \mathcal{X}} \text{ReLU} \big( \frac{1}{M} \sum_{j=1}^M \mathcal{B}_\varphi(f(x_i, \Delta_i, w_j), t) \\
& \qquad\qquad - \kappa \, \mathcal{B}_\varphi(x_i, t-1) - b_3 + \epsilon_M \big).
\end{aligned}
\tag{17}
$$

The loss $\mathcal{L}$ has five terms, each corresponding to a BC requirement in Definition 2.6. The first term (weight $c_\rho$) penalizes violations of the scalar feasibility constraint (10). The second term (weight $c_n$) enforces nonnegativity of $\mathcal{B}_\varphi$ over the sampled set $\mathcal{X}$. The third and fourth terms (weights $c_\mathcal{I}$ and $c_\mathcal{U}$) enforce the initial and terminal separation constraints (7) and (8). The last term (weight $c_f$) enforces condition (9) using an empirical mean over $M$ noise realizations and a slack $\epsilon_M > 0$ to account for empirical mean approximation. Training minimizes $\mathcal{L}$ to achieve $\mathcal{L} = 0$, which implies that $\mathcal{B}_\varphi$ and its associated constants satisfy all constraints on the sample sets $\mathcal{I}, \mathcal{U}$, and $\mathcal{X}$. The resulting certified radius is denoted by $\delta_{\text{cert}}$ (resp. $\delta_{\text{cert}}'$), with the learned constants $b_1^*, b_2^*, b_3^*, \kappa^*, \epsilon_M^*$.

### 3.3. Surrogate-Assisted Certificate Synthesis

The NNBC synthesis procedure requires parameter trajectories under different perturbation budgets and stochastic realizations of the training process. Repeatedly generating such rollouts from the true update map $f$ in (5) can be computationally expensive, since each rollout executes the underlying stochastic training pipeline. To improve scalability, we optionally employ a lightweight surrogate transition model $\hat{f}$ to generate additional trajectories for synthesizing a candidate NNBC. The surrogate is fitted from one-step transition pairs sampled from true rollouts,

$$(x(t), \Delta, t) \longmapsto x(t+1), \tag{18}$$

and it approximates the induced parameter-update behavior,

$$x(t+1) \approx \hat{f}(x(t), \Delta, t). \tag{19}$$

Thus, $\hat{f}$ serves only as a computational proxy for trajectory generation: it is neither a replacement for the classifier $h_x$ nor a model of the adversarial mechanism, but only approximates the input–output behavior of the training update map to enlarge the finite synthesis dataset.

The surrogate-generated trajectories are used solely for synthesizing a candidate $\mathcal{B}_\varphi$ via the loss in (17). Hence, $\hat{f}$ has a purely algorithmic role: it improves scalability and trajectory coverage during synthesis, but does not itself certify robustness. Regardless of whether the synthesis data are generated from $f$ or $\hat{f}$, the learned $\mathcal{B}_\varphi$ satisfies the constraints in (17) only on finitely many training samples. The next section therefore introduces a validation stage that extends the guarantee beyond these samples with some confidence.

# 4. Certificate Verification

The synthesis procedure in Section 3 yields an NNBC $\mathcal{B}_\varphi$ that meets conditions (7)–(10) on finitely many data. However, this does not imply that these inequalities hold beyond the sampled data, even within the proposed certified robust radii. This section closes this gap by developing a framework that verifies whether the candidate certificate $\mathcal{B}_\varphi$ satisfies inequalities (7)–(10) for all admissible perturbations. We further derive a *probably approximately correct* (PAC) guarantee with explicit confidence, quantifying the reliability of the certified robust radius beyond the training samples. To this end, we fix the trained NNBC $\mathcal{B}_\varphi$, the learned constants $b_1^*, b_2^*, b_3^*, \kappa^*, \epsilon_M^*$, and the candidate radii $\delta_{\text{cert}}$ (train-time) or $\delta'_{\text{cert}}$ (test-time). To assess the satisfaction of conditions (7)–(10) beyond the sampled data, we introduce auxiliary scalars $\eta_{k'} \in \mathbb{R}$ and thresholds $\tilde{b}_{k'} := b_{k'}^* + \eta_{k'}$, where $k' \in [3]$. Let $\bar{\Omega} \in \{\Omega, \Omega'\}$ denote the general admissible perturbation set, then, the constraint functions $q_k : \mathbb{R}^d \times \bar{\Omega} \times \mathbb{N}_{0:T} \to \mathbb{R}$, $\forall k \in [5]$, corresponding to the inequalities in Definition 2.6 are defined:

$$q_1(x,\Delta,t) = \left(-\mathcal{B}_\varphi^*(x,t)\right)\mathbf{1}_X, \tag{20}$$

$$q_2(x,\Delta,t) = \left(\mathcal{B}_\varphi^*(x,0) - \tilde{b}_1\right)\mathbf{1}_{X_0}, \tag{21}$$

$$q_3(x,\Delta,t) = \left(-\mathcal{B}_\varphi^*(x,T) + \tilde{b}_2\right)\mathbf{1}_{X_u^\alpha}, \tag{22}$$

$$q_4(x,\Delta,t) = \left(E(x,\Delta,t) - \kappa^*\mathcal{B}_\varphi^*(x,t\text{--}1) - \tilde{b}_3 + \epsilon_M^*\right)\mathbf{1}_X, \tag{23}$$

$$q_5(x,\Delta,t) = \tilde{b}_1\kappa^{*T} + \tilde{b}_3 T - \tilde{b}_2(1 - \rho), \tag{24}$$

where $E(x,\Delta,t) := \mathbb{E}\left[\mathcal{B}_\varphi^*(f(x,\Delta,w),t) \mid x, \Delta\right]$.

## 4.1. Robust Convex Problem (RCP)

To validate the BC conditions (7)–(10), we formulate the following RCP over the corresponding joint space $X \times \bar{\Omega}$ in terms of the scalar margins $\eta_{k'}$, $k' \in [3]$:

$$\text{RCP}: \begin{cases} \min_{\eta_{k'}} & \eta_1 + \eta_3 - \eta_2 \\ \text{s.t. } & q_k(x,\Delta,t) \leq 0, \\ & \forall(x,\Delta) \in (X \times \bar{\Omega}), \forall t \in \mathbb{N}_{0:T}, \\ & \tilde{b}_{k'} = b_{k'}^* + \eta_{k'}, \tilde{b}_{k'} \in \mathbb{R}_{>0}, \\ & \forall k \in [5], \forall k' \in [3], \end{cases} \tag{25}$$

where $\Delta \in \bar{\Omega}$ denote a generic admissible perturbation. The RCP determines the minimal margins $\eta_{k'}$ (with optimal values $\eta_{k'}^*$ for $k' \in [3]$) for which all constraints are satisfied

for every state $x \in X$ and for every admissible perturbation within the certified robust radii. This guarantees that the resulting certificate $\mathcal{B}_\varphi$ complies with Definition 2.6. To tackle the expectation term $E(x,\Delta,t)$ in (23), we substitute it with its empirical , i.e.,

$$\bar{E}(x,\Delta,t) := \frac{1}{M}\sum_{j=1}^M \mathcal{B}_\varphi\left(f(x, \Delta, w_j),t\right) + \epsilon_M^* \tag{26}$$

derived from $M$ independent and identically distributed (i.i.d.) samples of $w$, along with the approximation error $\epsilon_M^*$. Taking into account a confidence level of at least $1 - \beta_0 \in (0, 1]$, substituting $E$ by $\bar{E}$ into $q_4$ leads to the following relaxed RCP:

$$\text{RCP}_M: \begin{cases} \min_{\eta_{r,k'}} \eta_{r,1} + \eta_{r,3} - \eta_{r,2} \\ \text{s.t. } \bar{q}_k(x,\Delta,t) \leq 0, \\ \quad \forall(x, \Delta) \in (X \times \bar{\Omega}), \forall t \in \mathbb{N}_{0:T}, \\ \quad \tilde{b}_{k'} = b_{k'}^* + \eta_{r,k'}, \ \tilde{b}_{k'} \in \mathbb{R}_{>0}, \\ \quad \forall k \in [5], \ \forall k' \in [3], \\ \quad \text{where } \bar{q}_1 = q_1, \bar{q}_2 = q_2, \bar{q}_3 = q_3, \\ \quad \bar{q}_4 = \bar{E}(x,\Delta,t) - \kappa^*\mathcal{B}_\varphi(x,t\text{--}1) - \tilde{b}_3, \\ \quad \bar{q}_5 = \tilde{b}_1\kappa^{*T} + \tilde{b}_3 T - \tilde{b}_2\left(1 - \frac{\rho}{1-\beta_0}\right). \end{cases} \tag{27}$$

The subsequent lemma establishes a connection between the feasible solution of RCP in (25) and $\text{RCP}_M$ in (27).

**Lemma 4.1** ($\text{RCP}_M \to \text{RCP}$). *Consider an RCP in (25) and its relaxation $\text{RCP}_M$ in (27), where the expectation term in $q_4$ is replaced by its empirical mean over $M$ i.i.d. samples and its error $\epsilon_M^*$, and $q_5$ is adjusted to enforce a confidence level of at least $1 - \beta_0 \in (0, 1]$. Let $\mathcal{B}_\varphi$ be a fixed trained NNBC and let $\eta_{r,k'}$, $\forall k' \in [3]$, be a feasible solution of $\text{RCP}_M$. Assume there exists a unified $V_m > 0$ such that $\text{Var}[\mathcal{B}(f(x, \Delta, w),t) \mid x, \Delta] \leq V_m$ for all $(x, \Delta) \in (X \times \bar{\Omega})$ and all $t \in \mathbb{N}_{1:T}$. Then, with a confidence of at least $1 - \beta_0$, $\eta_{r,k'}$, $\forall k' \in [3]$, is also feasible for RCP provided that*

$$M \geq \frac{V_m}{\beta_0(\epsilon_M^*)^2}. \tag{28}$$

Note that solving $\text{RCP}_M$ is computationally intractable due to implicit dynamics $f$. To overcome this difficulty, we reformulate $\text{RCP}_M$ as a chance-constrained program (CCP).

## 4.2. Chance-Constrained Program (CCP)

In the CCP, the conditions in $\text{RCP}_M$ are required to hold with a prescribed probability rather than deterministically. Specifically, for a given violation level $v \in (0, 1)$, we introduce the aggregate constraint function $q_{\max} := \max_{k \in [5], t \in \mathbb{N}_{0:T}} \bar{q}_k(x, \Delta, t)$ and define

$$\text{CCP}: \begin{cases} \min_{\eta_{c,k'}} \eta_{c,1} + \eta_{c,3} - \eta_{c,2} \\ \text{s.t. } \mathbb{P}\left[q_{\max} \leq 0\right] \geq 1 - v, \\ \quad \tilde{b}_{k'} = b_{k'}^* + \eta_{c,k'}, \ \tilde{b}_{k'} \in \mathbb{R}_{>0}, \\ \quad \forall k \in [5], \ \forall k' \in [3], \end{cases} \tag{29}$$

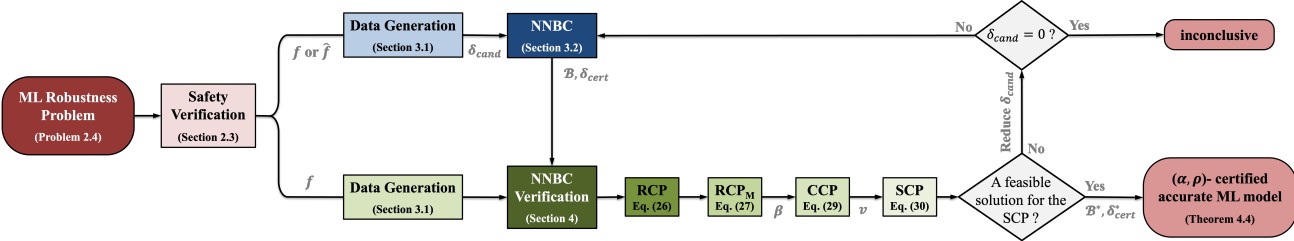

*Figure 2.* Certification workflow. The ML robustness problem is reformulated as a safety verification task. The NNBC is synthesized for a candidate radius $\delta_{\text{cand}}$ using sampled trajectories from either the true dynamics $f$ or, for scalable synthesis, a surrogate model $\hat{f}$. The resulting candidate certificate is then formally validated on fresh i.i.d. scenarios generated from the true dynamics $f$, by relaxing an intractable RCP through a series of approximations (RCP$_M$, CCP) to a tractable SCP. If there exists a feasible solution for the SCP, then with a confidence of at least $1 - \beta$, the ML model is $(\alpha, \rho)$-certified accurate with a violation probability at most $v$. Otherwise, $\delta_{\text{cand}}$ is reduced and the synthesis–verification loop repeats until either feasibility is achieved or $\delta_{\text{cand}} \to 0$, in which case the certification outcome is declared inconclusive (See Appendix C).

where the probability is considered across the random outcomes of the joint distribution of $(x, \Delta)$. Our objective is to address the CCP in (29) instead of the RCP$_M$ in (27). In the CCP, a subset of constraints with total probability mass up to $v$ is effectively discarded, thereby trading off robustness against feasibility. However, solving the CCP is challenging because both the state $x$ and the disturbance $\Delta$ belong to continuous domains. Consequently, we approximate the CCP by means of a scenario-based convex problem (SCP).

### 4.3. Scenario Convex Problem (SCP)

To render the CCP computationally feasible, we replace the infinitely many constraints with $N_s$ i.i.d. parameter samples drawn from the original training dynamics $f$ associated with the primary ML model $h_x$. These samples are obtained using the data-generation procedure described in Section 3.1 and are generated independently of the trajectories employed to train $\mathcal{B}_\varphi$. This process yields joint sampled sets $\mathcal{I}' \subseteq X_0 \times \bar{\Omega}$, $\mathcal{U}' \subseteq X_u^\alpha \times \bar{\Omega}$, and $\mathcal{X}' \subseteq X \times \bar{\Omega}$. Subsequently, SCP imposes the inequalities solely on these scenarios, $\forall i \in [N_s]$, (30):

$$
\text{SCP}: \begin{cases}
\min_{\eta_{s,k'}} \ \eta_{s,1} + \eta_{s,3} - \eta_{s,2} \\
\text{s.t. } \ \bar{q}_k(x_i, \Delta_i, t) \leq 0, \\
\quad \forall (x_i, \Delta_i) \in \mathcal{Z}_k, \ \forall t \in \mathbb{N}_{0:T}, \\
\quad \mathcal{Z}_1 = \mathcal{Z}_4 = \mathcal{X}', \ \mathcal{Z}_2 = \mathcal{I}', \ \mathcal{Z}_3 = \mathcal{U}', \\
\quad \tilde{b}_{k'} = b_{k'}^* + \eta_{s,k'}, \ \tilde{b}_{k'} \in \mathbb{R}_{>0}, \\
\quad \forall k \in [5], \ \forall k' \in [3].
\end{cases}
\tag{30}
$$

Let $\eta_{s,k'}^*$ represent the optimal margins derived from the SCP solution, resulting in the validated NNBC denoted by $\mathcal{B}_\varphi^*$ and its associated certified robust radius, represented by $\delta_{\text{cert}}^*$ (resp. $\delta_{\text{cert}}'^*$). These margins define the adjusted thresholds $\tilde{b}_{s,k'}$, certifying that the trained NNBC $\mathcal{B}_\varphi$ satisfies all the conditions in Definition 2.6 over the sample scenarios. However, since the SCP replaces the infinite set with finitely many data points, we establish a probabilistic bound that quantifies the gap between $\eta_{s,k'}^*$ and $\eta_{r,k'}^*$. To formalize the rigorous connection between the RCP, its CCP, and SCP,

we borrow the following lemma from (Calafiore & Campi, 2006), adapted to our setting.

**Lemma 4.2** (SCP → CCP). *Consider the CCP in (29) and its corresponding SCP in (30). Let $v, \beta \in (0, 1]$ be fixed constants. Then, for each $k' \in [3]$, the optimal solution $\eta_{s,k'}^*$ obtained from the SCP fulfills all RCP$_M$ constraints with a violation probability no larger than $v$, provided that the number of i.i.d. sampled scenarios $N_s$ satisfies*

$$
N_s \ \geq \ \left\lceil \frac{2}{v} \ln \frac{1}{\beta} \ + \ 6 \ + \ \frac{6}{v} \ln \frac{2}{v} \right\rceil.
\tag{31}
$$

Lemmas 4.1 and 4.2 provide the two probabilistic links needed for certification: the former connects the empirical expectation approximation to the RCP constraints, and the latter connects the finite SCP solution to the CCP. We now combine them with Theorem 2.8 to obtain the final certification guarantee.

### 4.4. Formal Probabilistic Guarantees

We now state the main certification result. Figure 2 summarizes the full synthesis–verification workflow, and Algorithm 1 presents its train-time implementation. The test-time case follows analogously, with perturbations entering through the evaluation set rather than the training dynamics.

**Theorem 4.3.** *Consider an ML model $h_x$, trained on a possibly poisoned dataset $\mathcal{D}_{\text{tr}}^\Delta$ and evaluated on a potentially perturbed test set $\mathcal{D}_{\text{te}}^{\Delta'}$, as described in Section 2.1. Let $\alpha, \rho, v, \beta \in (0, 1]$ be fixed constants, and assume that an NNBC $\mathcal{B}_\varphi^*$ is trained using a finite collection of sampled trajectories as in Section 3.1. If the number of i.i.d. samples $M$ used to approximate the expectation term and the number of i.i.d. scenarios $N_s$ employed to solve the SCP in (30) meet the conditions of Lemma 4.1 and Lemma 4.2, respectively, and the SCP admits a feasible solution, then $\mathcal{B}_\varphi^*$ is a valid NNBC and defines a certified robust radius $\delta_{\text{cert}}^*$ (respectively, $\delta_{\text{cert}}'^*$). Furthermore, with confidence at least $1 - \beta$, the resulting model is $(\alpha, \rho)$-certified accurate, with violation probability at most $v$ for all perturbations contained within this certified radius.*

---

**Algorithm 1**: Probabilistic Robustness Certification for Neural Networks against Poisoning Attacks

**Input**: Model $h_x$, thresholds $\alpha, \rho$, perturbation grid $\delta_{\text{grid}}$ (resp. $\delta'_{\text{grid}}$), trajectory budget $N_b$, surrogate-fitting budget $N_{\text{app}}$, empirical-mean samples $M$, SCP scenarios $N_s$, horizon $T$, violation level $v$, confidence parameters $\beta_0, \beta$, and reduction rule Reduce($\cdot$).
**Output**: Certified NNBC $\mathcal{B}^*_\varphi$ and certified radius $\delta^*_{\text{cert}}$ (resp. $\delta'^*_{\text{cert}}$), if certification succeeds; otherwise, an inconclusive outcome.
 1: Generate $N_b$ stochastic trajectories of the dt-SDS in (5) over $\delta_{\text{grid}}$ (resp. $\delta'_{\text{grid}}$), as described in Section 3.1.
 2: Evaluate terminal states using $\mathcal{G}$ in (3) and construct $\mathcal{I}, \mathcal{S}, \mathcal{U}$, and $\mathcal{X}$ as in (11)–(14).
 3: Estimate $\delta_{\text{emp}}$ (resp. $\delta'_{\text{emp}}$) using the empirical $(\alpha, \rho)$-accuracy criterion in (15) (resp. (16)).
 4: Initialize $\delta_{\text{cand}} \leftarrow \delta_{\text{emp}}$ (resp. $\delta'_{\text{cand}} \leftarrow \delta'_{\text{emp}}$) and certified $\leftarrow$ false.
 5: **while** $\delta_{\text{cand}} > 0$ (resp. $\delta'_{\text{cand}} > 0$) **do**
 6:     Restrict the synthesis data to rollouts satisfying $\|\Delta_i\|_p \le \delta_{\text{cand}}$ (resp. $\|\Delta'_i\|_p \le \delta'_{\text{cand}}$), consistent with Definition 2.1.
 7:     Fit the surrogate transition model $\hat{f}$ from $N_{\text{app}}$ transition pairs sampled from the selected true rollouts, as in Section 3.3.
 8:     Use $\hat{f}$ only for synthesis: generate additional rollouts and augment the current sampled sets $(\mathcal{I}, \mathcal{U}, \mathcal{X})$.
 9:     For each $(x_i, t) \in \mathcal{X}$, generate $M$ one-step successors to approximate the expectation term in the BC condition (9).
10:     Train $\mathcal{B}_\varphi$ by minimizing the NNBC loss in (17), obtaining $b_1^*, b_2^*, b_3^*, \kappa^*, \epsilon_M^*$.
11:     Fix $\mathcal{B}_\varphi$ and verify it using $N_s$ fresh i.i.d. scenarios generated from the true dynamics $f$, as in Section 4.
12:     Solve the SCP in (30) to obtain margins $\eta^*_{s,k'}, k' \in [3]$.
13:     **if** the SCP is feasible **then**
14:         Set $\mathcal{B}^*_\varphi \leftarrow \mathcal{B}_\varphi$, $\delta^*_{\text{cert}} \leftarrow \delta_{\text{cand}}$ (resp. $\delta'^*_{\text{cert}} \leftarrow \delta'_{\text{cand}}$), and certified $\leftarrow$ true.
15:         Break.
16:     **else**
17:         Set $\delta_{\text{cand}} \leftarrow$ Reduce($\delta_{\text{cand}}$) (resp. $\delta'_{\text{cand}} \leftarrow$ Reduce($\delta'_{\text{cand}}$)), following the synthesis–verification loop in Figure 2.
18:     **end if**
19: **end while**
20: If certified = false, declare the certification outcome inconclusive.

**Certification statement:** If certified = true and the sample sizes satisfy Lemmas 4.1 and 4.2, then, with confidence at least $1 - \beta$, $h_x$ is $(\alpha, \rho)$-certified accurate for all perturbations bounded by $\delta^*_{\text{cert}}$ (resp. $\delta'^*_{\text{cert}}$), with violation probability at most $v$.

---

*Remark* 4.4 (Verification Outcome). The SCP in (30) serves as the final computationally tractable validation step for the synthesized NNBC described in Section 3, and it leads to two possible outcomes. (i) If the SCP is feasible and has an optimal solution $\eta^*_{s,k'}$, then the candidate radius $\delta_{\text{cert}}$ (respectively, $\delta'_{\text{cert}}$) is confirmed as the certified radius $\delta^*_{\text{cert}}$ (respectively, $\delta'^*_{\text{cert}}$) (cf. Theorem 4.3). (ii) If the SCP is infeasible, then the trained NNBC violates at least one BC inequality over the sampled scenarios. In this situation (see Figure 2), we decrease $\delta_{\text{cand}}$ and iterate the synthesis/verification procedure until a feasible solution is found. If infeasibility persists even as $\delta_{\text{cand}} \to 0$ (despite retraining the NNBC), the verification step is deemed inconclusive.

## 5. Experimental Results

In this section, we evaluate the proposed framework and report certified train-time robustness of the learned classifier $h_x$ under $\ell_\infty$ and $\ell_2$ threat models. Additional test-time certification results, full configurations, and further comparisons are deferred to Appendix D.

**Experimental setup.** We consider three standard image-classification benchmarks: **MNIST**, **SVHN**, and **CIFAR-10**. At train time, we evaluate three representative poisoning strategies: Projected Gradient Descent (PGD) (Madry et al., 2018), Backdoor Attack (BDA) (Gu et al., 2017), and Bulls-eye Polytope Attack (BPA) (Aghakhani et al., 2021). At test time, we consider PGD and AutoAttack (AA) (Croce & Hein, 2020). The hypothesis class $h_x$ includes **MLP**, **CNN**, and **ResNet** models trained with **GD**, **SGD**, and **Adam**. Unless stated otherwise, stochasticity arises from random

initialization and optimizer-induced randomness, such as mini-batch sampling. Full hyperparameter settings and architecture specifications are reported in Appendix D.

**Certification results.** Table 1 reports representative train-time certification results across datasets, poisoning ratios $\tau \in [0.1, 1]$, and $\ell_\infty/\ell_2$-bounded attacks. Each reported value of $\delta^*_{\text{cert}}$ should be interpreted as a formally validated perturbation radius: for all admissible perturbations within this radius, the learned classifier is certified to satisfy the $(\alpha, \rho)$-accuracy requirement with confidence at least $1 - \beta = 99.9\%$, up to violation probability $v$. In all reported settings, the empirical-mean approximation uses $\beta_0 = 10^{-4}$, and the table lists the corresponding sample sizes $(N_b, M, N_s)$ used for NNBC synthesis, expectation approximation, and SCP validation. Figure 3 further illustrates the relationship between the empirical radius, the certified radius obtained by the main pipeline, and the certified radius obtained when NNBC synthesis is accelerated using surrogate-generated rollouts. As expected, $\delta^*_{\text{cert}}$ is generally smaller than $\delta_{\text{emp}}$: the latter is a sample-dependent estimate computed from observed stochastic rollouts, whereas the former is accepted only after the candidate NNBC passes SCP-based validation on fresh scenarios drawn from the true dynamics $f$. This gap reflects the price of moving from empirical robustness to a distribution-level PAC certificate over stochastic training outcomes. The surrogate-assisted certificates exhibit an additional, but controlled, conservatism due to the approximation gap between $\hat{f}$ and $f$ during synthesis; however, the final guarantee is still determined exclusively by verification under the true dynamics $f$.

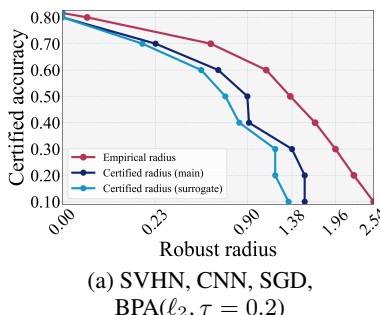

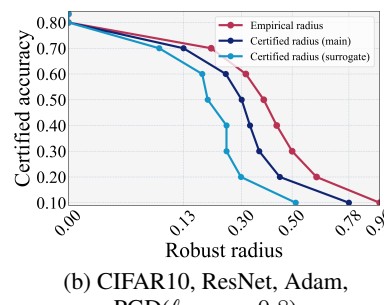

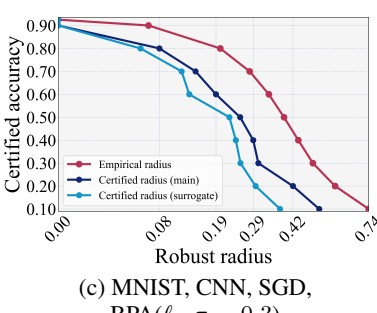

| (a) SVHN, CNN, SGD, BPA($\ell_2, \tau = 0.2$) | (b) CIFAR10, ResNet, Adam, PGD($\ell_\infty, \tau = 0.8$) | (c) MNIST, CNN, SGD, BPA($\ell_2, \tau = 0.3$) |
|---|---|---|

*Figure 3.* Certified accuracy ($\alpha$) versus perturbation magnitude ($\delta$) across poisoning scenarios. The NNBC is synthesized using trajectories from $f$ (main) or $\hat{f}$ (surrogate), and validated on fresh i.i.d. samples generated from $f$. Each subplot shows the empirical robust radius and the certified radius obtained by our framework with $\rho = 0.95$, $\beta = 10^{-3}$. Violation probabilities $v$ are: (a) 0.05, (b) 0.07, (c) 0.03.

*Table 1.* Train-time certification results across multiple settings. Each row reports the empirical ($\delta_{\text{emp}}$) and certified ($\delta_{\text{cert}}^*$) robustness radii for models that are ($\alpha, \rho$)-certified accurate with confidence $\geq 99.9\%$ and violation probability $\leq v$.

| Dataset | Model | $\mathcal{A}$ | $\ell_p$ | $N_b$ | $M$ | $N_s$ | $\alpha$ | $\rho$ | $\delta_{\text{emp}}$ | $\delta_{\text{cert}}^*$ | $v$ |
|---|---|---|---|---|---|---|---|---|---|---|---|
| MNIST | MLP | BPA | $\ell_\infty$ | 3000 | 300 | 1200 | 0.95 0.90 | 0.89 0.90 | 0.20 0.28 | 0.14 0.21 | 0.03 |
| SVHN | MLP | PGD | $\ell_2$ | 3000 | 200 | 1100 | 0.80 0.75 | 0.90 0.90 | 1.24 3.01 | 0.99 2.34 | 0.03 |
| | ResNet | BDA | $\ell_\infty$ | 2000 | 100 | 600 | 0.75 0.60 | 0.95 0.95 | 0.16 0.26 | 0.14 0.22 | 0.07 |
| CIFAR-10 | CNN | BPA | $\ell_\infty$ | 1500 | 100 | 500 | 0.75 0.70 | 0.90 0.90 | 0.32 0.58 | 0.21 0.45 | 0.08 |

*Table 2.* Comparison of certified robust radii between our method and RAB under test-time backdoor attacks (BDA) for CNN models trained with SGD.

| Dataset | $\tau$-ratio | $\ell_p$ | $\alpha$ | $\delta_{\text{emp}}$ | $\delta_{\text{cert}}^*$ (RAB) | $\delta_{\text{cert}}^*$ (Ours) |
|---|---|---|---|---|---|---|
| MNIST | 0.15 | $\ell_\infty$ | 0.90 | 0.08 | NA | 0.04 |
| | | | 0.80 | 0.16 | 0.10 | 0.11 |
| | | | 0.60 | 0.22 | 0.14 | 0.16 |
| SVHN | 0.10 | $\ell_2$ | 0.80 | 0.19 | NA | 0.12 |
| | | | 0.60 | 0.32 | 0.17 | 0.28 |
| | | | 0.40 | 0.66 | 0.28 | 0.52 |
| CIFAR-10 | 0.10 | $\ell_\infty$ | 0.50 | 0.05 | NA | 0.01 |
| | | | 0.40 | 0.09 | NA | 0.06 |
| | | | 0.30 | 0.15 | 0.05 | 0.09 |

**Comparison and interpretation.** Direct comparison with prior defenses is inherently delicate because existing certificates often target different objects: a single trained model, a deterministic training abstraction, or a threat model that is not norm-bounded in the same sense. The closest barrier-based framework is (Taheri et al., 2026), which also certifies parameter-space safety via barrier certificates but does not model training as a stochastic process. Its deterministic abstraction can therefore yield larger certified radii, while our certificates are deliberately more conservative because they account for the distribution of terminal parameters induced by stochastic training. We also compare with RAB (Weber et al., 2023), a randomized-smoothing-based certification method, in Table 2 and Figure 4. Across the reported regimes with $\rho \in [0.9, 0.96]$, our method certifies larger radii in several settings and continues to certify at accuracy thresholds where RAB returns "NA". These gains should be interpreted together with the broader scope of the proposed framework: the same NNBC–SCP verification principle applies to both train-time poisoning and test-time evasion, and does not require specifying the attack algorithm beyond the admissible perturbation set. We omit direct comparisons to defenses such as BagFlip (Zhang et al., 2022) and ensemble-based approaches (Levine & Feizi, 2021), since their corruption models or certification targets do not match the norm-bounded train/test-time setting considered here.

Overall, these results demonstrate that the proposed NNBC–SCP pipeline can produce non-trivial certified radii across datasets, architectures, optimizers, and attack types, while preserving a unified probabilistic interpretation under stochastic training.

# 6. Conclusion and Future Work

In this paper, we introduced a novel data-driven framework for certifying the robustness of ML models against train-time poisoning and test-time evasion attacks. Our core methodology models the stochastic training process as a *discrete-time stochastic dynamical system* (dt-SDS) and reformulates *robustness certification* as a formal *safety verification problem*. To overcome analytical intractability, we proposed an approach for synthesizing a *neural network barrier certificate* (NNBC) from sampled trajectories and then verifying the candidate BC by solving a scenario convex problem (SCP)—a tractable relaxation of the intractable robust convex problem (RCP). We then provide a *formal probabilistic guarantee* that the final performance of the ML model remains above a target threshold, despite stochastic training and bounded adversaries. Experiments demonstrate our ability to certify non-trivial perturbation budgets. The framework is notably *model-agnostic* and *attack-agnostic*, requiring *no prior knowledge of the adversary*, thereby highlighting its generality. Future work will address the applicability of the framework to other perturbation types, such as label or mixed feature-label corruptions, to enhance its utility in more complex adversarial settings.

## Impact Statement

Our work provides a formal framework for assessing neural network robustness against both train-time poisoning and test-time evasion under stochastic training. Although such insights could, in principle, be misused, we believe that identifying and understanding these vulnerabilities is essential for safe deployment of neural networks. The societal benefits of advancing robustness certification therefore outweigh the potential risks, and we do not anticipate any immediate misuse arising from this work. This paper was written entirely by the authors.

## Acknowledgments

This work is supported by the ConVeY DFG research training group and partially by NSF grants CNS-2145184 and CNS-2111688.

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

## A. Notation

We denote the set of real numbers by $\mathbb{R}$, the set of non-negative real numbers by $\mathbb{R}_{\geq 0}$, the set of non-positive real numbers by $\mathbb{R}_{\leq 0}$, the set of strictly positive real numbers by $\mathbb{R}_{>0}$, and the set of strictly negative real numbers by $\mathbb{R}_{<0}$. The set of positive integers is denoted by $\mathbb{N}$, and we define $\mathbb{N}_0 := \{0\} \cup \mathbb{N}$. For any integers $a, b \in \mathbb{N}_0$ with $a \leq b$, we define the index set $\mathbb{N}_{a:b} := \{a, a+1, \ldots, b\}$. For a scalar $x \in \mathbb{R}$, we write $|x|$ for its absolute value and $\lceil x \rceil$ for its ceiling. The $d$-dimensional Euclidean space is denoted by $\mathbb{R}^d$. For any $x \in \mathbb{R}^d$ and $p \geq 1$, we use the $\ell_p$-norm $\|x\|_p := (\sum_{i=1}^d |x_i|^p)^{1/p}$; in particular, $\|x\|_\infty := \max_{1 \leq i \leq d} |x_i|$. The rectified linear unit (ReLU) is defined pointwise as $\mathrm{ReLU}(x) := \max\{0, x\}$. For a set $X \subseteq \mathbb{R}^d$, the indicator function $\mathbf{1}_X : \mathbb{R}^d \to \{0, 1\}$ is defined by $\mathbf{1}_X(x) = 1$ if $x \in X$, and $0$ otherwise. The complement of a set $A \subseteq B$ within a universal set $B$ is denoted by $B \backslash A$. For $r \in \mathbb{N}$, we use the shorthand $[r] := \{1, \ldots, r\}$. We denote expectation and probability (with respect to the probability space under consideration) by $\mathbb{E}[\cdot]$ and $\mathbb{P}(\cdot)$, respectively. The variance of a random variable $z$ is defined by $\mathrm{Var}(z) = \mathbb{E}(z^2) - (\mathbb{E}(z))^2$, where $\mathbb{E}(z)$ is its expected value. For any finite set $A$, the notation $|A|$ denotes its cardinality, i.e., the number of distinct elements contained in $A$. The logical negation operator (NOT) is denoted by $\neg$.

## B. Proofs

### B.1. Proof of Theorem 2.8

*Proof.* Let $\mathcal{B}(x, t)$ be a BC satisfying the conditions in Definition 2.6 with constants $b_1, b_2, b_3 \in \mathbb{R}_{>0}$ and $\kappa \in (0, 1]$. Let $x(t)$ denote the state (model parameters) at iteration $t$, resulting from the stochastic dynamics in (5), starting from an arbitrary initial state $x(0) \in X_0$. We aim to find an upper bound on the probability of the final state $x(T)$ being in the unsafe set, i.e., $\mathbb{P}[x(T) \in X_u^\alpha \mid x(0)]$.

Let $e(t) := \mathbb{E}[\mathcal{B}(x(t), t) \mid x(0)]$ be the expected value of the barrier function at time $t$, conditioned on the *specific* starting state $x(0)$. The base case is at $t = 0$:

$$e(0) = \mathbb{E}[\mathcal{B}(x(0), 0) \mid x(0)] = \mathcal{B}(x(0), 0) \leq b_1, \tag{32}$$

where the inequality holds from condition (7), since $x(0) \in X_0$. For the recursive step ($t \in \mathbb{N}_{1:T}$), one obtains:

$$e(t) = \mathbb{E}[\mathcal{B}(x(t), t) \mid x(0)] \tag{33}$$

$$= \mathbb{E}\big[\mathbb{E}[\mathcal{B}(x(t), t) \mid x(t-1), x(0)] \mid x(0)\big] \tag{34}$$

$$= \mathbb{E}\big[\mathbb{E}[\mathcal{B}(x(t), t) \mid x(t-1)] \mid x(0)\big] \tag{35}$$

In (34) and (35), we applied the tower property of conditional expectations and the Markov property of system dynamics, where the subsequent state $x(t)$ is dependent solely on the preceding state $x(t-1)$. Then, by applying condition (9), one has,

$$e(t) \leq \mathbb{E}\big[\kappa \mathcal{B}(x(t-1), t-1) + b_3 \mid x(0)\big] \tag{36}$$

$$= \kappa \, \mathbb{E}[\mathcal{B}(x(t-1), t-1) \mid x(0)] + b_3 \tag{37}$$

$$= \kappa \, e(t-1) + b_3. \tag{38}$$

We proceed by expanding the recursive inequality from (38) over $T$ iterations, beginning with the condition $e(0) \leq b_1$:

$$e(T) \leq \kappa e(T-1) + b_3 \tag{39}$$

$$\leq \kappa \big(\kappa e(T-2) + b_3\big) + b_3 = \kappa^2 e(T-2) + b_3(1 + \kappa)$$

$$\vdots$$

$$\leq \kappa^T e(0) + b_3 \sum_{i=0}^{T-1} \kappa^i \leq b_1 \kappa^T + b_3 \sum_{i=0}^{T-1} \kappa^i \leq b_1 \kappa^T + b_3 T, \tag{40}$$

where the last inequality is the result of $\kappa \in (0, 1]$. Hence, one obtains:

$$\mathbb{E}[\mathcal{B}(x(T), T) \mid x(0)] \leq b_1 \kappa^T + b_3 T. \tag{41}$$

Now, we relate this expected value to the probability of reaching the unsafe set $X_u^\alpha$. By condition (8), any terminal state $x(T) \in X_u^\alpha$ must satisfy $\mathcal{B}(x(T), T) \geq b_2$. This implies that the event $\{x(T) \in X_u^\alpha\}$ is a subset of the event $\{\mathcal{B}(x(T), T) \geq b_2\}$. Thus, for any $x(0) \in X_0$:

$$\mathbb{P}[x(T) \in X_u^\alpha \mid x(0)] \leq \mathbb{P}[\mathcal{B}(x(T), T) \geq b_2 \mid x(0)]. \tag{42}$$

By applying Markov's inequality to the non-negative random variable $\mathcal{B}(x(T), T)$ and its threshold $b_2 > 0$, we obtain:

$$\mathbb{P}[\mathcal{B}(x(T), T) \geq b_2 \mid x(0)] \leq \frac{\mathbb{E}[\mathcal{B}(x(T), T) \mid x(0)]}{b_2}. \tag{43}$$

By combining the last two inequalities and substituting the bound from (41), one obtains:

$$\mathbb{P}[x(T) \in X_u^\alpha \mid x(0)] \leq \frac{b_1 \kappa^T + b_3 T}{b_2}. \tag{44}$$

The probability of being *safe* (i.e., $x(T) \in X_s^\alpha$, which is equivalent to $\mathcal{G}(x(T)) \geq \alpha$) is given by:

$$\mathbb{P}[\mathcal{G}(x(T)) \geq \alpha \mid x(0)] = 1 - \mathbb{P}[x(T) \in X_u^\alpha \mid x(0)] \tag{45}$$
$$\geq 1 - \frac{b_1 \kappa^T + b_3 T}{b_2}.$$

Finally, the existence of the BC (Definition 2.6) requires condition (10) to hold, which is $b_1 \kappa^T + b_3 T \leq b_2(1 - \rho)$. Rearranging this gives:

$$\frac{b_1 \kappa^T + b_3 T}{b_2} \leq 1 - \rho \implies 1 - \frac{b_1 \kappa^T + b_3 T}{b_2} \geq \rho. \tag{46}$$

Therefore, $\mathbb{P}[\mathcal{G}(x(T)) \geq \alpha] \geq \rho$. This confirms that the ML model $h_x$ is $(\alpha, \rho)$-certified accurate for all specified perturbations, completing the proof. $\qquad\square$

### B.2. Proof of Lemma 4.1

*Proof.* Let $(\eta_{r,k'}, \tilde{b}_{k'})$ be an arbitrary feasible solution for the problem $\text{RCP}_M$ in (27). We will demonstrate that this solution is also feasible for the original RCP in (25), provided the sample size $M$ satisfies the condition in (28).

A feasible solution for the RCP must satisfy $q_k \leq 0$ for all $k \in \{1, \dots, 5\}$. First, for $k \in \{1, 2, 3\}$, the constraints are identical by definition, as $\bar{q}_k = q_k$. Since the solution is feasible for $\text{RCP}_M$, these constraints are necessarily satisfied.

The core of the proof lies in showing that the constraints $\bar{q}_4 \leq 0$ and $\bar{q}_5 \leq 0$ jointly imply the satisfaction of the original constraints $q_4 \leq 0$ and $q_5 \leq 0$. Let $\mu(x, \Delta, t) := \mathbb{E}[\mathcal{B}^*(f(x, \Delta, w), t) \mid x, \Delta]$ denote the true, unknown expectation and $\hat{\mu}_M(x, \Delta, t) := \frac{1}{M} \sum_{j=1}^M \mathcal{B}^*(f(x, \Delta, w_j), t)$ denote the empirical mean computed from $M$ i.i.d. samples. We use Chebyshev's inequality to relate these terms. The variance of the empirical mean is $\text{Var}(\hat{\mu}_M(x, \Delta, t)) = \frac{\text{Var}(\mathcal{B}^*(x^+, t) \mid x, \Delta)}{M}$. Given the lemma's assumption, i.e., $\text{Var}(\mathcal{B}(x^+, t) \mid x, \Delta) \leq V_m$, we have $\text{Var}(\hat{\mu}_M(x, \Delta, t)) \leq \frac{V_m}{M}$. Now, we apply Chebyshev's inequality (Hernández, 2001) to get the following for any $\epsilon_M^* > 0$:

$$\mathbb{P}\left(|\hat{\mu}_M(x, \Delta, t) - \mu(x, \Delta, t)| \geq \epsilon_M^*\right) \leq \frac{\text{Var}(\hat{\mu}_M(x, \Delta, t))}{(\epsilon_M^*)^2} \leq \frac{V_m}{M(\epsilon_M^*)^2}. \tag{47}$$

Note that, the probability in Chebyshev's inequality is taken only over the draw of $w_1, \dots, w_M$, for each fixed $(x, \Delta, t)$. By substituting the lemma's condition, $M \geq \frac{V_m}{\beta_0(\epsilon_M)^2}$, this probability is bounded by $\beta_0$:

$$\mathbb{P}\left(|\hat{\mu}_M(x, \Delta, t) - \mu(x, \Delta, t)| \geq \epsilon_M\right) \leq \beta_0. \tag{48}$$

This implies that the event $E_M := \{|\hat{\mu}_M(x, \Delta, t) - \mu(x, \Delta, t)| \leq \epsilon_M^*\}$ occurs with probability $\mathbb{P}(E_M) \geq 1 - \beta_0$. This event $E_M$ guarantees the one-sided bound $\mu(x, \Delta, t) \leq \hat{\mu}_M(x, \Delta, t) + \epsilon_M^*$, which is critical for our analysis. Since the solution $(\eta_{r,k'}, \tilde{b}_{k'})$ is feasible for $\text{RCP}_M$, the constraint $\bar{q}_4 \leq 0$ holds:

$$\hat{\mu}_M(x, \Delta, t) + \epsilon_M^* \leq \kappa \mathcal{B}(x, t-1) + b_3. \tag{49}$$

Conditional on the event $E_\mathrm{M}$ (which occurs with probability of at least $\geq 1 - \beta_0$), we combine the bound from Chebyshev's inequality with the feasible constraint to get:

$$\mu(x, \Delta, t) \leq \hat{\mu}_M(x, \Delta, t) + \epsilon_M^* \leq \kappa \mathcal{B}(x, t-1) + b_3, \tag{50}$$

This demonstrates that $q_4(x,t) = \mu(x, \Delta, t) - \kappa^* \mathcal{B}^*(x, t-1) - b_3 \leq 0$ holds conditionally on $E_\mathrm{M}$.

Finally, we show that $q_5 \leq 0$ holds. Let $E_\mathrm{safe}$ be the event $x(T) \in X_s^\alpha$. Using the law of total probability and the non-negativity of probability:

$$\begin{aligned} \mathbb{P}(E_\mathrm{safe}) &= \mathbb{P}(E_\mathrm{safe} \mid E_\mathrm{M})\mathbb{P}(E_\mathrm{M}) + \mathbb{P}(E_\mathrm{safe} \mid \neg E_\mathrm{M})\mathbb{P}(\neg E_\mathrm{M}) \\ &\geq \mathbb{P}(E_\mathrm{safe} \mid E_\mathrm{M})\mathbb{P}(E_\mathrm{M}) \geq \mathbb{P}(E_\mathrm{safe} \mid E_\mathrm{M})(1 - \beta_0). \end{aligned} \tag{51}$$

Since $E_\mathrm{M}$ implies $q_4 \leq 0$ (and $q_1, q_2, q_3$ are satisfied), all BC conditions hold conditionally on $E_\mathrm{M}$. We can therefore bound the conditional probability $P' := \mathbb{P}(E_\mathrm{safe} \mid E_\mathrm{M})$:

$$P' \geq 1 - \frac{\tilde{b}_1 \kappa^{*T} + \tilde{b}_3 T}{\tilde{b}_2}. \tag{52}$$

Substituting this into our bound for $\mathbb{P}(E_\mathrm{safe})$ gives:

$$\mathbb{P}(E_\mathrm{safe}) \geq \left(1 - \frac{\tilde{b}_1 \kappa^{*T} + \tilde{b}_3 T}{\tilde{b}_2}\right)(1 - \beta_0). \tag{53}$$

We now use the last feasibility condition, $\bar{q}_5 \leq 0$:

$$\bar{q}_5 = \tilde{b}_1 \kappa^{*T} + \tilde{b}_3 T - \tilde{b}_2 \left(1 - \frac{\rho}{1 - \beta_0}\right) \leq 0. \tag{54}$$

Rearranging the previous inequality yields the bound for the conditional safety term:

$$\begin{aligned} \tilde{b}_1 \kappa^{*T} + \tilde{b}_3 T &\leq \tilde{b}_2 \left(1 - \frac{\rho}{1 - \beta_0}\right) \\ \implies 1 - \frac{\tilde{b}_1 \kappa^{*T} + \tilde{b}_3 T}{\tilde{b}_2} &\geq \frac{\rho}{1 - \beta_0}. \end{aligned} \tag{55}$$

Substituting this result back into the bound for $\mathbb{P}(E_\mathrm{safe})$:

$$\mathbb{P}(E_\mathrm{safe}) \geq \left(1 - \frac{\tilde{b}_1 \kappa^{*T} + \tilde{b}_3 T}{\tilde{b}_2}\right)(1 - \beta_0) \geq \rho. \tag{56}$$

Hence, We have $\mathbb{P}(E_\mathrm{safe}) \geq \rho$. This implies the satisfaction of the original global constraint $q_5 = \tilde{b}_1 \kappa^{*T} + \tilde{b}_3 T - \tilde{b}_2(1 - \rho) \leq 0$. Since the feasible solution $(\eta_{r,k'}, \tilde{b}_{k'})$ satisfies $q_k \leq 0$ for all $k \in \{1, \ldots, 5\}$, it is, therefore, a feasible solution for the original RCP. $\square$

### B.3. Proof of Lemma 4.2

*Proof.* Let $\mathbb{P}$ represent a probability measure in the product space $X \times \Omega$, where $X$ is the parameter space of the ML model, and $\Omega$ represents the collection of permissible poisoning perturbation matrices $\Delta$. Since solving the $\mathrm{RCP}_M$ is generally intractable, we relax it to a chance-constrained problem (CCP). The CCP in (29) requires that the constraints $\bar{q}_k(x, t) \leq 0$ hold with probability at least $1 - v$ for all $k \in [5]$. To quantify the fraction of the uncertainty space where a solution $\eta$ violates any constraint, we define the violation probability function:

$$\mathbb{V}(\eta) := \mathbb{P}\left[(x, \Delta) \in X \times \Omega : \exists k \in [5] \text{ s.t. } \max_{k \in [5]} \bar{q}_k(x, \Delta, t) \geq 0\right]. \tag{57}$$

Hence, a solution $\eta$ is said to be $v$-*feasible* if $\mathbb{V}(\eta) \leq v$, i.e., it satisfies all CCP constraints with a probability of at least $1 - v$. Solving the CCP exactly is still intractable due to its infinite constraint set over the continuous set $X \times \Omega$. The SCP in (30) circumvents this by enforcing the constraints only on a finite number $N_s$ of i.i.d. samples from $\mathbb{P}$. Let $\eta_{s,k'}^*$ denote the optimal SCP solution. According to the scenario approach theory (Calafiore & Campi, 2006; Campi & Garatti, 2008), if the SCP has a support rank $r$ (the number of constraints that can become active and determine the solution), then the probability that the obtained $\eta_{s,k'}^*$ violates the original CCP constraints beyond the level $v$ is bounded as

$$\mathbb{P}^{N_s}\left(\mathbb{V}(\eta_{s,k'}^*) > v\right) \ \leq \ \sum_{j=0}^{r-1} \binom{N_s}{j} v^j (1-v)^{N_s-j}, \tag{58}$$

where $\mathbb{P}^{\hat{N}} = \mathbb{P} \times \cdots \times \mathbb{P}$ (taken $\hat{N}$ times) denotes the product measure over the $N_s$ independent samples. To ensure that the probability of such a violation does not exceed $\beta$, we require

$$\sum_{j=0}^{r-1} \binom{N_s}{j} v^j (1-v)^{N_s-j} \ \leq \ \beta. \tag{59}$$

Following Corollary 1 in (Calafiore & Campi, 2006), a closed-form conservative bound that guarantees this inequality is given by:

$$N_s \ \geq \ \left\lceil \frac{2}{v} \ln\frac{1}{\beta} \ + \ 2r \ + \ \frac{2r}{v} \ln\frac{2}{v} \right\rceil. \tag{60}$$

Since the BC $\mathcal{B}_\varphi$ is trained and fixed, the scalar decision variables $\eta_{s,k'}$ are the only decision variables in the optimization problem; thus, the support rank $r$ is at most 3. Substituting $r = 3$ yields (31). Therefore, if the number of sampled scenarios satisfies (31), then with confidence at least $1 - \beta$, the SCP solution $\eta_{s,k'}^*$ is $v$-feasible for the CCP and thus satisfies the constraints of the original RCP$_M$ with a violation probability of at most $v$. $\qquad\square$

### B.4. Proof of Theorem 4.3

*Proof.* The proof forms a direct chain from the sample-based SCP to the final certification claim. Solving the SCP in (30) yields an optimal margin $\eta_{s,k'}^*$. By Lemma 4.2, with $N_s$ scenarios, this solution is $v$-feasible for the CCP (29) with confidence at least $1 - \beta$, meaning the constraints $\bar{q}_k$ are violated with a probability of at most $v$. Lemma 4.1 shows that the problem RCP$_M$ is a conservative relaxation of the ideal problem RCP (25). Thus, any violation of the ideal constraints $q_k$ would also violate the constraints $\bar{q}_k$, so the $v$-feasibility guarantee transfers to the ideal constraints. Since the ideal constraints $q_k$ coincide with the BC conditions in Definition 2.6, Theorem 2.8 implies that satisfying them ensures the model $h_x$ is $(\alpha, \rho)$-certified accurate, with a violation probability of at most $v$ and a confidence of at least $1 - \beta$, completing the proof. $\qquad\square$

## C. Scalable Workflow

Our framework follows the workflow in Figure 2 and is organized into two main components: *synthesis* and *verification*. In this appendix, we provide a brief, self-contained description of how these two components interact to produce a certified robust radius, emphasizing the practical choices that enable a scalable implementation under computational constraints. (For more details, check: `https://figshare.com/s/42f69e5af3c98213688c`.)

### C.1. Synthesis

The NNBC synthesis in Section 3 requires a trajectory dataset of parameter iterates obtained by rolling out the training dynamics under multiple perturbation budgets. In principle, these trajectories are generated from the true update map $x(t+1) = f(x(t), \Delta, w(t))$. However, for large-scale models, repeatedly producing rollouts under $f$ (i.e., retraining across many $\delta'$ and initializations) becomes the dominant computational bottleneck. To enable scalable synthesis, our framework optionally allows replacing true rollouts by rollouts from a lightweight surrogate update model $\hat{f}$, which is used *only* to produce additional trajectories for learning a *candidate* NNBC. Concretely, we first collect a limited set of true rollouts under $f$ (as in Section 3.1), yielding supervised pairs $(x(t), \Delta, t) \mapsto x(t+1)$, and fit $\hat{f}$ via $x(t + 1) \approx \hat{f}(x(t), \Delta, t)$. In the stochastic setting, $\hat{f}$ may be viewed as approximating a typical update induced by $w(t)$ (e.g., a conditional mean), but our guarantees do not rely on any modeling assumption about $\hat{f}$. The role of $\hat{f}$ is purely algorithmic: it trades trajectory-generation cost for synthesis speed, and the user can choose between $f$ and $\hat{f}$ depending on available hardware and the

desired fidelity/conservatism trade-off. Since the parameter dimension is large, we fit $\hat{f}$ in a low-dimensional latent space obtained via *Principal Component Analysis (PCA)* on collected iterates. Let $P \in \mathbb{R}^{d \times r}$ contain the top-$r$ principal directions and define $z(t) = P^\top x(t) \in \mathbb{R}^r$. We then learn and roll out the surrogate in latent coordinates, and decode via $\hat{x}(t) = Pz(t)$. A key practical point is that the effect of perturbation on the parameter trajectory is not necessarily linear or monotone in $\Delta$: different attack strategies, hyperparameters, and training settings can change which directions in parameter space become active. Therefore, selecting $r$ based on a single perturbation level can be misleading. Instead, before fixing $r$, we inspect the cumulative explained variance across trajectories generated under multiple $\delta'$, and choose the smallest $r$ that preserves a target variance level *uniformly* over the considered budgets (e.g., the minimal $r$ achieving $95\% - 99\%$ explained variance across all plotted $\delta'$). This yields a principled, setting-aware compression that informs the surrogate design while maintaining scalability.

### C.2. Verification

Crucially, the surrogate model $\hat{f}$ is never used in the verification stage. After training a candidate NNBC $\mathcal{B}_\varphi$ (possibly using surrogate-generated trajectories), we fix $\mathcal{B}_\varphi$ and certify robustness by solving the SCP in (30) using *fresh i.i.d scenarios generated from the true dynamics $f$*. Hence, all PAC guarantees in Theorem 4.3 are stated *with respect to $f$*. A poor surrogate can only affect synthesis efficiency (e.g., by producing candidates that fail SCP or require more iterations), but it cannot produce an incorrectly certified radius, since certification is accepted only if the fixed certificate passes verification under $f$.

## D. Additional Results and Simulation Details

This appendix complements the main experimental section by reporting the full set of configurations and results that underlie our certification claims. Our end-to-end workflow is summarized in Figure 2: for each dataset/model/attack setting, we (i) generate stochastic training trajectories (optionally using the scalable surrogate workflow in Appendix C for NNBC *synthesis*), (ii) train a candidate NNBC $\mathcal{B}_\varphi$ via the loss in (17), and (iii) validate the fixed NNBC on *fresh* i.i.d scenarios generated from the true dynamics $f$ by solving the SCP in (30). This synthesis/ verification loop returns the certified robust radius $\delta_{\text{cert}}^*$ (or $\delta_{\text{cert}}'^*$ for test-time), together with a PAC-style confidence and a bound on the violation probability.

### D.1. Results

The full certification curves are shown in Figure 5, and all per-setting radii and PAC quantities are reported in Table 3. Figure 5 illustrates the trade-off between the accuracy threshold $\alpha$ and the admissible perturbation level $\delta$: red curves show the *empirical* robust radius $\delta_{\text{emp}}$ estimated from sampled trajectories, while green curves show the *certified* robust radius $\delta_{\text{cert}}^*$ returned by our synthesis/ verification pipeline. Here, $\delta_{\text{emp}}$ is the largest grid value for which sampled rollouts remain safe, whereas $\delta_{\text{cert}}^*$ is the formally validated radius obtained via SCP-based PAC validation. Complete configurations for Table 1 and Figure 5 are summarized in Table 4 (datasets, attacks, optimizers, models, and NNBC settings). The **Attack** block lists the attack type (PGD, BPA, BDA, AA), the $p$-norm, poisoning ratio, and step size; **ML Setup** specifies the model, optimizer, and learning rate; and **Certificate** reports the NNBC training and the resulting PAC bounds on the violation probability $v$ at confidence $1 - \beta$. **Runtime** is reported in minutes.

In addition, the NNBC $\mathcal{B}_\varphi$ is trained via the loss function in (17) using Adam, with balanced loss weights normalized by the cardinality of the generation sets. Figure 5 illustrates the trade-off between the accuracy threshold $\alpha$ and the robust radius $\delta$. The red curves represent the *empirical robust radius* ($\delta_{\text{emp}}$)—the maximum perturbation where sampled trajectories remain safe. The green curves represent the *certified robust radius* ($\delta_{\text{cert}}^*$), which is the formal guarantee computed by our framework. The NNBC architecture choices used across all experiments are listed in Table 4 (depth/optimizer), and the underlying classifier architectures are summarized in Table 5, where **Conv Layers** report the number of convolutional layers and their output channels and **Params (M)** provides the approximate number of trainable parameters (in millions).

### D.2. Comparisons to prior works.

Figure 4 and Table 2 compare our certificates with two representative baselines under a matched protocol (same dataset/model and attack-time setting, sweeping the same range of accuracy thresholds $\alpha$). In general, our comparisons highlight the importance of explicitly accounting for stochastic training effects. Methods such as RAB certify robustness for a *single trained model instance* and therefore provide guarantees that are conditional on that particular realization of training, without modeling the randomness inherent in modern pipelines.

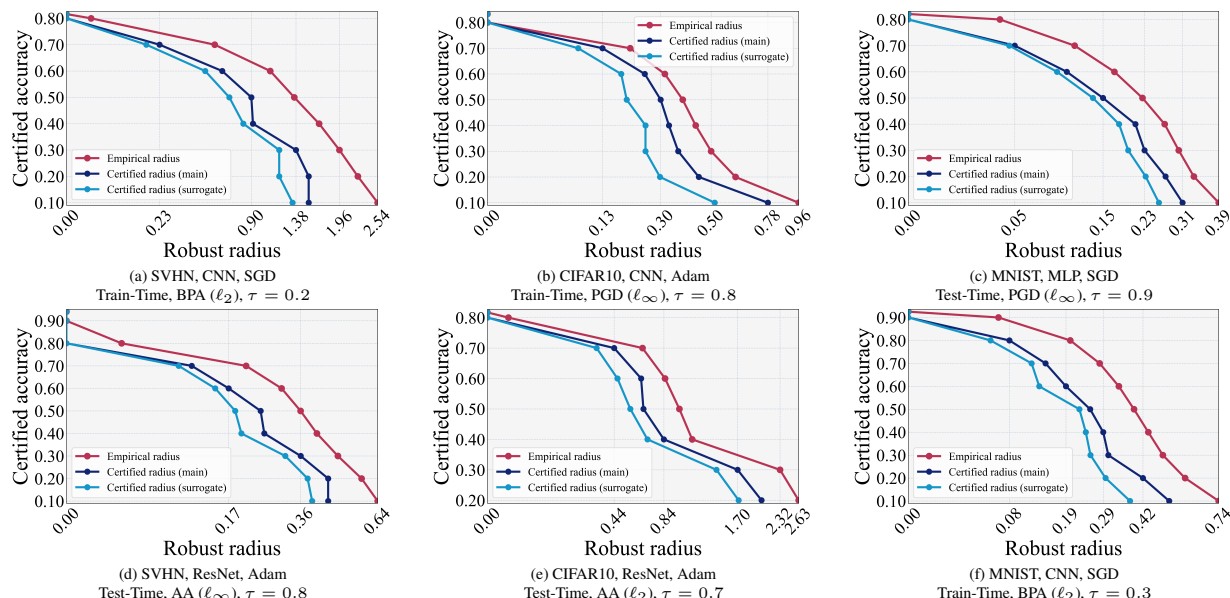

*Figure 5.* Certified accuracy ($\alpha$) versus perturbation magnitude ($\delta$) across poisoning scenarios. The NNBC is synthesized using trajectories from $f$ (main) or $\hat{f}$ (surrogate), and validated on fresh i.i.d. samples generated from $f$. Each subplot shows the empirical robust radius and the certified radius obtained by our framework with $\rho = 0.95$, $\beta = 10^{-3}$. Violation probabilities $v$ are: (a) 0.05, (b) 0.07, (c) 0.02, (d) 0.06, (e) 0.08, (f) 0.03.

In contrast, we model learning as a dt-SDS and derive probabilistic guarantees with respect to the training-induced randomness, namely over the distribution of terminal parameters produced by the stochastic process. This difference helps explain why our certificates remain meaningful in regimes where stochasticity materially affects robustness, while instance-conditional certificates can become conservative or fail to certify (Table 2). Moreover, our formulation is not tied to a specific attack mechanism within the bounded-perturbation threat model, and it covers both poisoning and evasion attacks under the same verification principle, whereas RAB targets a more specialized setting. In addition, compared to (Taheri et al., 2026), the key distinction is the training model. While both approaches certify parameter-space safety via barrier functions, (Taheri et al., 2026) adopts a deterministic abstraction that effectively collapses the pipeline into a fixed update

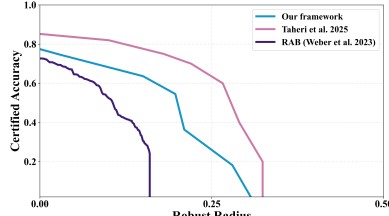

*Figure 4.* Comparison of our framework and RAB results on SVHN under test-time BDA with $\ell_\infty$ attack.

map, thereby ignoring training-time stochasticity. This modeling choice can yield systematically larger certified radii, consistent with its curves lying above ours. Our framework instead treats training as a dt-SDS (Definition 2.4) and certifies robustness under stochastic dynamics, producing radii that may be more conservative but remain valid across repeated executions of the same nominal configuration.

*Table 3.* Certification results across multiple ML models and attack configurations. Each row reports the empirical ($\delta_{\mathrm{emp}}$) and certified ($\delta^*_{\mathrm{cert}}$) robustness radii obtained for ML models that are ($\alpha, \rho$)-certified accurate, with a confidence level of at least 99.9% and a violation probability at most $v$.

| Dataset | ML model | Optimizer | $\mathcal{A}$ | Attack Time | $\tau$-ratio | $\ell_p$ | $N_b$ | M | $N_s$ | $\alpha$ | $\rho$ | $\delta_{\mathrm{emp}}$ | $\delta^*_{\mathrm{cert}}$ | $v$ |
|---|---|---|---|---|---|---|---|---|---|---|---|---|---|---|
| MNIST | MLP | SGD | BPA | Train-Time | 0.3 | $\ell_\infty$ | 3000 | 300 | 1200 | 0.95 / 0.90 | 0.89 / 0.90 | 0.20 / 0.28 | 0.14 / 0.21 | 0.03 |
| | CNN | Adam | BDA | Test-Time | 0.1 | $\ell_2$ | 2500 | 200 | 1000 | 0.90 / 0.80 | 0.96 / 0.97 | 0.31 / 1.04 | 0.29 / 0.96 | 0.04 |
| SVHN | MLP | SGD | PGD | Train-Time | 0.9 | $\ell_2$ | 3000 | 200 | 1100 | 0.80 / 0.75 | 0.90 / 0.90 | 1.24 / 3.01 | 0.99 / 2.34 | 0.03 |
| | CNN | SGD | AA | Test-Time | 0.8 | $\ell_\infty$ | 2000 | 150 | 650 | 0.85 / 0.80 | 0.92 / 0.94 | 0.08 / 0.15 | 0.06 / 0.09 | 0.07 |
| | ResNet | Adam | BDA | Train-Time | 0.2 | $\ell_\infty$ | 2000 | 100 | 600 | 0.75 / 0.60 | 0.95 / 0.95 | 0.16 / 0.26 | 0.14 / 0.22 | 0.07 |
| CIFAR-10 | CNN | Adam | BPA | Train-Time | 0.1 | $\ell_\infty$ | 1500 | 100 | 500 | 0.75 / 0.70 | 0.90 / 0.90 | 0.32 / 0.58 | 0.21 / 0.45 | 0.08 |
| | ResNet | Adam | PGD | Test-Time | 0.6 | $\ell_2$ | 1000 | 50 | 400 | 0.75 / 0.60 | 0.96 / 0.95 | 0.08 / 0.89 | 0.06 / 0.76 | 0.09 |

*Table 4.* Experimental configurations across datasets, attacks, optimizers, models, and NNBC settings. **Attack** lists the attack type, $\ell_p$-norm, attack ratio $\tau$, and step size; **ML Setup** specifies the model and optimizer (learning rate in parentheses). **Certificate** reports the certification pipeline: **Surrogate model** includes $N_{app}$ (number of samples used to fit the approximation) and the surrogate model $\hat{f}$; **(PCs)** is the number of retained principal components. **NNBC** gives the NNBC training configuration ($N_b$, **M**, depth, optimizer), and **PAC** reports $N_s$ and the violation level $v$. **Runtime** (minutes) reports **Main** and **Surrogate**; the parenthetical **cost** is the percentage drop in certified radius incurred by the surrogate relative to the main pipeline.

| Dataset | Attack | | | | ML Setup | | | Certificate | | | | | | | | | Runtime | | Fig/Table |
|---|---|---|---|---|---|---|---|---|---|---|---|---|---|---|---|---|---|---|
| | | | | | | | | Surrogate model | | NNBC | | | | PAC | | Main | Surrogate | |
| | Type | $\ell_p$ | $\tau$ | Step | Model | Optimizer ($l_r$) | Type | $N_{app}$ | model (PCs) | $N_b$ | M | Layer | Optimizer ($l_r$) | $N_s$ | $v$ | t | t (cost) | |
| MNIST | PGD | $\ell_\infty$ | 0.9 | 40 | MLP | SGD (0.01) | Test-Time | 600 | MLP (15) | 3000 | 350 | 7 | Adam (0.001) | 1400 | 0.02 | 122 | 4 (-18%) | Fig 5(c) |
| MNIST | BPA | $\ell_2$ | 0.3 | 30 | CNN | SGD (0.01) | Train-Time | 400 | MLP (25) | 2200 | 250 | 5 | Adam (0.001) | 1000 | 0.03 | 156 | 7 (-29%) | Fig 5(f) |
| MNIST | BPA | $\ell_\infty$ | 0.3 | 30 | MLP | SGD (0.01) | Train-Time | – | – | 3000 | 300 | 8 | Adam (0.001) | 1200 | 0.03 | 114 | – | Table 1 |
| MNIST | BDA | $\ell_2$ | 0.1 | 30 | CNN | Adam (0.10) | Test-Time | – | – | 2500 | 200 | 5 | Adam (0.001) | 1000 | 0.03 | 151 | – | Table 1 |
| SVHN | BPA | $\ell_2$ | 0.2 | 10 | CNN | SGD (0.01) | Train-Time | 400 | MLP (30) | 2000 | 150 | 8 | Adam (0.001) | 700 | 0.05 | 176 | 11 (-14%) | Fig 5(a) |
| SVHN | AA | $\ell_\infty$ | 0.8 | 30 | ResNet | Adam (0.001) | Test-Time | 250 | MLP (50) | 1500 | 120 | 6 | Adam (0.001) | 550 | 0.06 | 231 | 15 (-9%) | Fig 5(d) |
| SVHN | PGD | $\ell_2$ | 0.9 | 40 | MLP | SGD (0.01) | Train-Time | – | – | 3000 | 200 | 7 | Adam (0.001) | 1100 | 0.03 | 136 | – | Table 1 |
| SVHN | AA | $\ell_\infty$ | 0.8 | 100 | CNN | SGD (0.01) | Test-Time | – | – | 2000 | 150 | 6 | Adam (0.001) | 650 | 0.06 | 182 | – | Table 1 |
| SVHN | BDA | $\ell_2$ | 0.2 | 30 | ResNet | Adam (0.001) | Train-Time | – | – | 2000 | 100 | 6 | Adam (0.001) | 600 | 0.06 | 194 | – | Table 1 |
| CIFAR10 | PGD | $\ell_\infty$ | 0.8 | 40 | CNN | Adam (0.001) | Train-Time | 450 | MLP (60) | 2000 | 150 | 9 | Adam (0.001) | 500 | 0.07 | 249 | 16 (-35%) | Fig 5(b) |
| CIFAR10 | AA | $\ell_2$ | 0.7 | 100 | ResNet | Adam (0.001) | Test-Time | 400 | MLP (65) | 1500 | 80 | 9 | Adam (0.001) | 450 | 0.08 | 263 | 21 (-22%) | Fig 5(e) |
| CIFAR10 | BPA | $\ell_\infty$ | 0.1 | 30 | CNN | Adam (0.001) | Train-Time | – | – | 1500 | 100 | 9 | Adam (0.001) | 500 | 0.08 | 218 | – | Table 1 |
| CIFAR10 | PGD | $\ell_2$ | 0.6 | 40 | ResNet | Adam (0.001) | Test-Time | – | – | 1000 | 50 | 7 | Adam (0.001) | 400 | 0.09 | 237 | – | Table 1 |

*Table 5.* Architectural specifications of models used across datasets. **Conv Layers** reports the number of convolutional layers and their output channels. **Pooling** specifies the type and frequency of downsampling. **FC Layers** denotes the fully connected layers with hidden dimensions up to the output layer. **Params (M)** provides the approximate number of trainable parameters (in millions).

| Dataset | Model | Conv Layers | Params (M) |
|---|---|---|---|
| MNIST | CNN | 3 conv (32, 64, 128) | ∼1.2M |
| | MLP | – | ∼0.6M |
| | LeNet | 2 conv (16, 32) | ∼0.1M |
| SVHN | CNN | 2 conv (64, 128) | ∼1.5M |
| | MLP | – | ∼3.2M |
| | ResNet18 | 18 conv (standard) | ∼11M |
| CIFAR-10 | CNN | 3 conv (64, 128, 256) | ∼4.5M |
| | ResNet18 | 18 conv (standard) | ∼11M |

# E. Discussion

**Fully agnostic framework.**   A central strength of our approach is that it is *fully agnostic* to the internal mechanics of modern ML pipelines. The certificate reasons only about the *distribution of parameter trajectories* induced by the *stochastic* training process. Concretely, we model learning as a dt-SDS $x(t+1) = f(x(t), \Delta, w(t))$ and express robustness as a *terminal safety* requirement over $X_s^\alpha$. All details of how training is implemented, how randomness enters the pipeline, and how perturbations are generated are absorbed into the black-box transition map $f$ and the stochastic input $w(t)$. As a result, our guarantee depends only on whether the barrier inequalities hold along trajectories produced by the true pipeline, and does not rely on an explicit model of $f$, on distributional assumptions about $w$, or on structural assumptions about the attack mechanism beyond the stated perturbation budget. This viewpoint unifies different robustness modalities under a single verification principle, since the certificate treats them through their common effect on the induced trajectory distribution. In this sense, the framework is agnostic by construction: it certifies the stochastic process through observed evolutions in parameter space, rather than any particular mechanism used to generate them.

**Robustness certificates for train-time vs. test-time.** We treat both train-time poisoning and test-time evasion within the same dt-SDS verification pipeline by casting robustness as terminal safety. For train-time poisoning, perturbations $\Delta$ enter the update rule $x^+ = f(x, \Delta, w)$ and reshape the distribution of trajectories $\xi$, and the BC certifies that $\mathbb{P}[\mathcal{G}(x(T)) \geq \alpha] \geq \rho$ uniformly over $\|\Delta\|_p \leq \delta_{\text{cert}}$. For test-time evasion, $f$ is unchanged but perturbations $\Delta'$ modify $\mathcal{G}$ and thus shift $(X_s^\alpha, X_u^\alpha)$; the same BC inequalities are verified uniformly over $\|\Delta'\|_p \leq \delta'_{\text{cert}}$. Hence, the difference is only where the adversary enters (dynamics vs. specification), while the certificate and verification machinery remain identical.

**Terminal versus trajectory certification.** Our barrier formulation in Definition 2.6 targets terminal safety, ensuring that the final parameters $x(T)$ satisfy the accuracy specification with some probability, while allowing intermediate iterates to transiently leave the safe set. This choice matches the ML objective—deployment depends on the terminal model—while avoiding unnecessary conservatism that would arise from requiring safety at every training step in high-dimensional nonconvex optimization. Extending the framework to full-trajectory guarantees is conceptually possible by redefining the unsafe set over the entire horizon, but would typically require stronger invariance conditions and more samples, and may substantially reduce certified radii.

**Influence of Data Generation and Sampling.** The effectiveness of our certification depends critically on the data generation process, involving three sampling stages: $N_b$ poisoned trajectories for learning the NNBC, $N_s$ i.i.d. scenarios for the SCP, and $M$ samples for expectation approximation. While $N_s$ is derived analytically via (31) based on the violation rate $v$, the choices of $N_b$ and $M$ are empirical. Our experiments indicate that reliable certification generally requires at least $N_b = 500$ and $M = 50$. Since the NNBC is learned from data, $N_b$ must be sufficiently large to capture the boundary between safe and unsafe regions induced by stochastic variability. Similarly, $M$ must be large enough to reduce approximation error. When NNBC training fails or the SCP is infeasible, we observed that incrementally increasing $N_b$ by $\sim 200$ and $M$ by $50$ restores feasibility, offering a practical balance between statistical coverage and computational cost.

**Effect of surrogate modeling in train-time vs. test-time certification.** A notable difference between train-time and test-time certification in our framework is how accurately a surrogate transition model can capture the underlying training dynamics. In the *test-time* setting, the training data remain clean and the perturbation level is effectively fixed at $\delta = 0$; repeated runs therefore expose the surrogate only to the intrinsic stochasticity of the pipeline (e.g., initialization, mini-batching, augmentation, optimizer noise). This yields a substantially simpler identification problem in which the surrogate can learn the stochastic variability of the update map with high fidelity from relatively few trajectories. In contrast, the *train-time* setting requires the surrogate to jointly model both sources of variability: the inherent stochasticity and the systematic effect of poisoning magnitude $\delta$ on the training evolution, which increases the effective complexity of the dynamics and typically demands richer trajectory coverage to achieve comparable approximation quality. Empirically, this asymmetry manifests in our certification curves: surrogate-based certificates in test-time are generally tighter and track the non-surrogate (main) certified radii more closely, whereas surrogate-based train-time certificates can be more conservative unless additional trajectories are used to resolve the $\delta$-dependent deformation of the training dynamics.

**Why a lightweight surrogate can still work for challenging models?** A natural question is how a single lightweight surrogate (e.g., an MLP in a PCA latent space) can remain effective across settings that use substantially different base learners, such as MLPs and ResNets. The key is that our surrogate is *not* a substitute for the underlying ML model, nor a model of the attack mechanism; rather, it is a lightweight proxy for the *training update map* viewed as an input–output operator that induces parameter trajectories. Once the full training pipeline (architecture, optimizer, data processing, stochasticity, and perturbations) is viewed as a black-box transition operator, the surrogate only needs to reproduce the coarse trajectory-level behavior relevant for synthesizing a *candidate* NNBC. Increasing the latent dimension (retaining more principal components) and making minor hyperparameter or architectural adjustments to the surrogate typically suffices to capture the dominant low-dimensional modes of parameter evolution, even when the underlying learner is more complex. Crucially, surrogate rollouts are used only to accelerate candidate construction; the certified radius is accepted only after PAC-style verification on fresh scenarios generated by the true dynamics. Therefore, an imperfect surrogate can at worst reduce synthesis efficiency or yield candidates that fail verification, but it cannot lead to a falsely certified guarantee: it either passes verification under the true pipeline or it is rejected.

**Limitations.** Our current framework is developed for the threat model introduced in Definition 2.1, namely input-space, label-preserving, fixed-ratio, $\ell_p$-bounded perturbations at train time or test time. Thus, while the framework is attack-agnostic with respect to the specific algorithm used to construct admissible perturbations within this threat model, it does not currently cover more general corruption classes such as label poisoning, mixed feature–label attacks, or unbounded/adaptive threat models outside this formulation. In addition, our certification target is defined with respect to a fixed held-out test dataset through $\mathcal{G}$ in (3), and therefore the resulting guarantee is an empirical benchmark-level certificate rather than a population-level generalization guarantee. Finally, the current implementation remains computationally demanding, since NNBC synthesis and SCP-based verification require repeated trajectory generation under stochastic training, which can limit scalability for larger models, longer horizons, or more complex datasets.

