# OpenReview forum: "Probabilistic Robustness Certificates against Adversarial Attacks"
_ICML.cc/2026/Conference — ICML 2026 regular_

### Official Review · Reviewer_2NVM · 2026-02-23

**Soundness:** 3
**Presentation:** 2
**Significance:** 3
**Originality:** 3
**Overall Recommendation:** 4
**Confidence:** 2

**Summary:**

This paper proposes a probabilistic framework for certifying ML model robustness against data poisoning attacks. The key idea is to model gradient-based training as a discrete-time stochastic dynamical system (dt-SDS) and formulate poisoning robustness as a probabilistic safety verification problem. The framework handles both train-time poisoning (perturbing training inputs within budget delta) and test-time evasion (perturbing test inputs within budget delta'), by defining the safe set in terms of model accuracy under test-time attacks. A barrier certificate (BC) provides sufficient conditions for the model parameters to stay within this safe set with probability at least rho. The BC is parameterized as a neural network and trained on sampled training trajectories. Exact certification is intractable, so the paper derives a tractable approximation. Theorem 4.4 gives a PAC guarantee: with confidence at least 1-beta the model is (alpha, rho)-certified accurate, given enough samples. Experiments on MNIST, SVHN, and CIFAR-10 with various architectures (MLP, CNN, ResNet) and attacks (PGD, BDA, Bullseye Polytope Attack) show that the method produces certified radii that are about 70-75% of the empirically observed radii, and outperforms RAB at higher accuracy thresholds where RAB often returns no certificate.

**Compliance With Llm Reviewing Policy:**

Affirmed.

**Final Justification:**

I thank the authors for the clarification. The distinction between checking a fixed trained model and certifying the stochastic training pipeline is clear, but for a deployed model the certificate still does not say whether that specific model is safe, which remains a limitation of the framework. The runtime numbers in Table 4 address A2.

**Key Questions For Authors:**

1. Does the framework still provide valid guarantees when the optimizer is non-Markovian, as in the case of Adam?
2. How do the runtimes scale as the number of model parameters increases?
3. Does training the barrier certificate on surrogate trajectories affect the PAC guarantees?

**Limitations:**

The paper does not include a limitations section. The method is restricted to lp-bounded poisoning of training inputs, as mentioned in the paper.

**Strengths And Weaknesses:**

Strengths:
1. The framing of gradient-based training as a dt-SDS and poisoning robustness as a safety verification problem is intuitive and clean. It also unifies poisoning and evasion robustness under a single framework.
2. The method provides formal PAC guarantees: Theorem 4.4 certifies that, with confidence at least 1-beta, the model is (alpha, rho)-certified accurate, given enough samples.
3. The approach is model-agnostic and attack-agnostic. It works across different optimizers (GD, SGD, Adam), architectures (MLP, CNN, ResNet), and attack types without modification.
4. In Table 2, the proposed approach achieves larger certified radii than RAB at higher accuracy thresholds (alpha = 0.8, 0.9), where RAB often returns NA.

Weaknesses:
1. The dt-SDS model assumes that the next state x(t+1) depends only on the current state x(t), i.e., training is Markovian. However, Adam maintains first and second moment estimates of past gradients, so the update at step t+1 depends on the full history of gradients, not just x(t). This means the Markov assumption does not hold for Adam. The paper includes Adam experiments but does not discuss this. Does the framework still provide valid guarantees when the optimizer is non-Markovian?
2. The guarantee is distributional: it says that over the randomness of training, at least rho fraction of runs produce a model with accuracy >= alpha. In practice, you train once and deploy that specific model. The certificate does not say whether that particular trained model is safe, only that most training runs would be. This is a weaker guarantee than per-model certification, and the paper does not discuss this distinction.
3. The framework requires sampling Nb poisoning trajectories, generating M next-state samples per state, and drawing Ns SCP samples. Table 4 reports some configuration details, but it is not clear what these numbers mean in terms of wall-clock time and how they scale with model size. Is the method practical for larger networks?
4. The barrier certificate is trained on surrogate trajectories rather than the true dynamics. Does training on a surrogate affect the PAC guarantees?

Soundness: I am not an expert in barrier certificates and scenario convex programs, but the theoretical framework seems sound.

Presentation: The paper is organized but the technical parts are not easy to follow for a reader who is not familiar with barrier certificates and scenario programs. It would help to add high-level explanations before definitions, lemmas, and theorems to give the reader intuition before the formal statements.

Significance: Certifying robustness of the training process against data poisoning, without assumptions on the model family or the attack strategy, is an important problem.

Originality: Modeling gradient-based training as a dt-SDS and using barrier certificates for poisoning robustness certification is new as far as I understand.

---

> ### Author Rebuttal · Authors · 2026-03-31
>
> We are grateful for the reviewer’s feedback and provide our responses below.
>
> Q1. Does the framework still provide valid guarantees when the optimizer is non-Markovian?
>
> A1. Yes. The key point is that our theory does not rely on Markovianity of the parameter process itself. The essential ingredient is the one-step conditional-expectation barrier inequality together with the tower property of conditional expectation; thus, what is required is only a well-defined conditional update with respect to the information available up to time $t-1$; see (5). This is particularly relevant for adaptive optimizers such as Adam. If the learning state is represented only by the model parameters $x(t)$, then the update depends implicitly on additional optimizer-memory variables, so the parameter process alone is generally not Markovian. However, this does not invalidate our framework, because the barrier argument is formulated at the level of conditional expectations rather than parameter-only Markovianity. Equivalently, one may augment the state to
> $
> \bar{x}(t):=\bigl(x(t),m(t),v(t)\bigr),
> $
> where $m(t)$ and $v(t)$ are Adam’s first- and second-moment estimates. Then the dynamics take the form
> $
> \bar{x}(t+1)=f\bigl(\bar{x}(t),\mathcal{D}_{\mathrm{tr}}^{\Delta}(t),w(t)\bigr),
> $
> which is first-order in the usual dynamical-systems sense. Hence, Adam-type memory does not invalidate the barrier conditions, the Chebyshev relaxation, or the scenario/PAC verification step. We will clarify this in the revised manuscript.
>
> Q2. How do runtimes scale as the number of model parameters increases? Is the method practical for larger networks?
>
> A2. We agree that Table 4 does not yet make the computational meaning of the reported configurations sufficiently explicit. Here, runtime refers to the end-to-end wall-clock time of the certification pipeline under a fixed experimental setting. As model size grows, each training rollout and verification step becomes more expensive, so the overall runtime increases accordingly. Thus, the main limitation for larger networks is computational rather than conceptual.
> More broadly, high runtime remains an open challenge across robustness-certification methods, which is precisely why our framework introduces surrogate-based synthesis. It is also important to compare runtime at the level of certification granularity. A large portion of the certification literature provides point-wise certificates, which require repeating the certification procedure for each individual test sample; their end-to-end computational cost therefore scales with the size of the test set. By contrast, the present method produces a single model-level certificate for the whole setting, and surrogate-based synthesis further reduces the computational burden. We agree that this was not stated clearly enough in the current manuscript, and we will make this runtime perspective more explicit in the revised version.
>
>
> Q3. Does training the barrier certificate on surrogate trajectories affect the PAC guarantees?
>
> A3. Training the NNBC on surrogate-generated trajectories can affect only the quality of the candidate certificate, not the validity of the final PAC guarantee. As clarified in Remark 3.1 and Appendix C, the surrogate model $\hat f$ is used only during synthesis. If $\hat f$ is inaccurate, the candidate may be inaccurate or conservative; this affects only verification, where the SCP in (25) is solved on fresh i.i.d. scenarios generated from the true dynamics $f$. In that case, the problem may become infeasible or certify a smaller radius, but no false guarantee is produced. Therefore, the PAC guarantee in Theorem 4.4 is unaffected by whether the candidate NNBC was synthesized using $f$ or $\hat f$, since it is derived exclusively from verification under the true dynamics $f$.
>
> W2. The guarantee is distributional over training randomness. In practice, one trains once and deploys a specific model. The certificate does not say whether that particular trained model is safe.
>
> A4. The reviewer is correct that our guarantee is not instance-specific for one realized trained model. This is intentional, because the certified object in our paper is the stochastic training procedure rather than one terminal model from a single run. This is also central to the framework: by certifying the learning process through sampled parameter trajectories, rather than one realized model or one test prediction, the method remains largely agnostic to architecture-specific structure and to the attack-generation mechanism. This is especially natural in our setting because training is stochastic, so repeated executions of the same nominal pipeline may lead to different terminal parameters. Accordingly, the result provides a distributional guarantee over the stochastic learning dynamics. Post hoc certification of one fixed trained model is a different certification target and lies outside the scope of the present paper.

---

> > ### Author Rebuttal · Reviewer_2NVM · 2026-04-01
> >
> > I thank the authors for their responses. A1 and A3 address my concerns. A4 is honest but remains a limitation: in practice, a user trains once and deploys that model, so a distributional guarantee over training runs does not tell them whether their specific deployed model is safe. Regarding A2, could the authors report wall-clock times for their current experiments and how these scale with model size?

---

> > > ### Author Response · Authors · 2026-04-05
> > >
> > > We thank the reviewer for this important follow-up observation.
> > >
> > > First, we believe it is useful to distinguish between certification of a single realized trained model and certification of the stochastic training pipeline in relation to the limitation you pointed out.
> > > In our paper, the performance functional is defined as
> > > \\[
> > > \mathcal{G}(x):=\frac{1}{n'}\sum_{i=1}^{n'} \mathbf{1} \left\\{ \arg\max_{j\in[k]} \left(h_x(u_i'+\delta_i') \right)_j = y_i' \right\\},
> > > \\]
> > >
> > > and the corresponding safe set is
> > > \\[
> > > X_s^\alpha := \\{x\in X \mid \mathcal{G}(x) \ge \alpha \\}.
> > > \\]
> > >
> > > Therefore, if one is given a specific fixed trained model with terminal parameter \\( x^\* \\), then its safety under our definition can be checked directly by evaluating \\( \mathcal{G}(x^\*) \\). In particular,
> > > \\[
> > > x^* \in X_s^\alpha
> > > \quad \Longleftrightarrow \quad
> > > \mathcal{G}(x^\*) \ge \alpha.
> > > \\]
> > >
> > > Thus, at the level of one realized trained model, determining whether it is safe is immediate in our framework and does not require the full certification machinery.
> > > However, this is different from the main certification question studied in the paper. Our notion of \\((\alpha,\rho)\\)-certified accuracy is
> > > \\[
> > > \mathbb{P} \left[ \mathcal{G}(x(T)) \ge \alpha \right] \ge \rho,
> > > \\]
> > >
> > > where the probability is taken over the randomness of the training procedure and hence over the terminal parameter \\( x(T) \\). Accordingly, the contribution of our framework is not merely to decide whether one fixed realization \\( x^\* \\) is safe, but to provide a formal probabilistic guarantee that the overall stochastic training-and-evaluation pipeline yields safe terminal models with probability at least \\( \rho \\), under admissible perturbations.
> > > We will clarify this distinction explicitly in the revised manuscript: a fixed realized model can be assessed directly through \\( \mathcal{G}(x^\*) \\), whereas the proposed certification framework is introduced to certify the reliability of the stochastic pipeline across realizations.
> > >
> > >
> > >
> > >
> > > Regarding A2, we thank the reviewer for pointing this out. We did report wall-clock times in the appendix: Table 4 includes runtime in minutes for all experiments, for both the main pipeline and the surrogate-based variant. Across the reported settings, the main runtime ranges from 114 to 263 minutes, while the surrogate-based variant ranges from 4 to 21 minutes when used. However, we agree that this may not have been sufficiently emphasized in the main discussion. We will revise the manuscript to highlight these numbers more clearly.
> > > Regarding scaling, we also agree that the current draft does not explicitly discuss this point in enough detail. Table 5 reports the approximate parameter counts of the evaluated models, ranging from about 0.1M to 11M parameters. The larger-model and larger-dataset settings, such as the ResNet18 and CIFAR-10 experiments, are generally among the most expensive runs. However, the runtime is not determined by the model size alone; it is also influenced by the certification hyperparameters, including the number of generated trajectories, the empirical expectation sample size, the number of SCP scenarios, the training horizon, and whether a surrogate model is used. We will add a brief discussion to make this dependence explicit in the revised version.
> > >
> > >
> > > We sincerely thank the reviewer for carefully reading our responses and for the valuable follow-up observation. This has been very helpful in improving the clarity and precision of the revised manuscript.

---

### Official Review · Reviewer_Nsqz · 2026-03-03

**Soundness:** 3
**Presentation:** 3
**Significance:** 2
**Originality:** 2
**Overall Recommendation:** 4
**Confidence:** 3

**Summary:**

The authors present a probabilistic framework for certifying the robustness of machine learning models against both train-time data poisoning and test-time evasion attacks. They begin by formally defining robustness: a model is considered robust if its final test accuracy remains probabilistically guaranteed to stay above a specific target threshold, even when subjected to worst-case adversarial perturbations within a bounded radius.To achieve this, overall, the authors focus on a central concept: modeling the inherently noisy gradient-based training pipeline (considering several sources of randomness) as a discrete-time stochastic dynamical system. Under this perspective, they reframe adversarial robustness as a formal probabilistic safety verification task.This authors propose a scalable, data-driven method to compute these guarantees using Neural Network Barrier Certificates. The NNBC is trained on a finite sample of stochastic training trajectories. This certificate learns to draw a strict boundary between "safe" parameter states and "unsafe" parameter states. Finally, they validate this learned certificate by solving a Scenario Convex Problem (SCP). This pipeline ultimately yields a PAC robustness bound. It provides a formally certified $\ell_p$-norm robust radius, mathematically guaranteeing (with a specified high confidence level) that the model will remain safe under attack. This is achieved in a model and attack-agnostic way.

**Compliance With Llm Reviewing Policy:**

Affirmed.

**Final Justification:**

I maintain my intial score. The authors correctly address my points. They recongnise the scalibility limitation if their approach and the fact that the guarantee provided is conditional on the particular held-out test set. Those are limitations that prevent me from raising my score. But overall, I think the paper is novel and sound.

**Key Questions For Authors:**

* Since the safe parameter boundary is anchored to a fixed empirical test set, how does statistical sampling variability affect the guarantee?

* What is the empirical failure rate of the verification? I.e. what proportion of the time does the proposed system fail to find a valid certificate (even after shrinking the radius) for a given model?

* As the model relies on generating thousands of full training trajectories... how well does it scale to much larger, modern architectures?

**Limitations:**

Yes

**Strengths And Weaknesses:**

* **Soundness**: The paper is mathematically sound, and the proofs appear correct. The theoretical assumptions are generally reasonable. To increase realism of the proposed verification method, I encourage the authors to discuss (maybe empirically) how frequently the SCP returns an "inconclusive" result.

* **Presentation**: The paper is well written and presented. However, plots (especially Fig. 3) are quite small and hard to read.

* **Significance**: The work is significant, as it addresses the problem of certifiable robustness in ML, which is key for the safe adoption of these technologies. The fact that the authors build certificates taking into account training randomness increases the significance of the work. However, there is a theoretical boundary regarding the practical scope of these guarantees that might be worth exploring or clarifying in the text. The definition of the "safe" and "unsafe" parameter sets is anchored to a fixed, empirical test set. Consequently, the formal PAC bounds strictly guarantee that the model will resist unseen attacks evaluated on this specific dataset. Thus, it seems this robustness is guaranteed modulo the statistical sampling variability of the test data. I recommend the authors include a brief discussion on this nuance to provide a fuller picture of the framework's scope.

* **Originality**: This work is incremental with respect to Taheri et al. (2025) ["Robustness Certificates for Neural Networks against Adversarial Attacks," arXiv:2512.20865]. The primary distinction between the two frameworks is that the current submission explicitly accounts for the stochasticity inherently present in the training process of machine learning models. Given this close relationship to prior work, the overall originality is moderate.

---

> ### Author Rebuttal · Authors · 2026-03-31
>
> We are grateful for the reviewer’s feedback and provide our responses below.
>
> Q1. Since the safe parameter boundary is anchored to a fixed empirical test set, how does statistical sampling variability affect the guarantee?
>
> A1. The reviewer is correct: because the safety specification is defined through the empirical performance functional $\mathcal{G}$ in (3), sampling variability in the held-out test set affects the guarantee through the safe/unsafe boundary itself. In particular, a terminal model is declared safe or unsafe according to whether its empirical performance on the chosen held-out set exceeds the threshold. Hence, if the held-out set is replaced by another sample from the same population, the value of $\mathcal{G}$ for the same terminal model may change, so the induced safe/unsafe partition and the certified radius may also change. Thus, the guarantee is conditional on the particular held-out test set and should be interpreted as a certificate relative to a fixed evaluation protocol, not as a population-level guarantee. A population-level statement would require an additional generalization layer relating $\mathcal{G}$ to its population counterpart. We will clarify this point more explicitly in the revised manuscript; extending the framework to account for test-set sampling variability is an important future direction, but outside the scope of the present paper.
>
> Q2. What is the empirical failure rate of the verification? What proportion of the time does the system fail to find a valid certificate (even after shrinking the radius)?
>
> A2. We agree that explicitly reporting the empirical inconclusive rate is useful. In our experiments, infeasibility at the SCP stage was not handled only by shrinking $\delta_{\mathrm{cand}}$: we first increased the amount of sampled trajectory data used for NNBC synthesis, and only then reduced $\delta_{\mathrm{cand}}$ if needed. When constructing the certified-accuracy curves in Fig. 5, certification was evaluated sequentially across target accuracies $\alpha$; if certification became inconclusive at a higher $\alpha$, we retained the last feasible radius from the preceding lower-$\alpha$ point only for plotting consistency. This explains why some consecutive accuracies in Fig. 5 share the same certified radius. Importantly, this applies only to the visualization: the numerical table results include only settings for which certification was actually obtained.
> Across the six settings in Fig. 5, corresponding to roughly 60 certification attempts, the empirical inconclusive rate was low overall: about $5\%$ (roughly 3/60) for the standard pipeline based on the true dynamics $f$, and about $8\%$ (roughly 5/60) for the surrogate-based pipeline, where $\hat f$ is used only during synthesis. We also note that this rate could often be reduced by increasing the admissible violation rate $v$; however, we deliberately kept both $1-\beta$ and $v$ fixed within each experimental setting to preserve a uniform verification protocol and keep the certified radii comparable along each curve. We will report this empirical inconclusive-rate analysis explicitly in the revised manuscript.
>
> Q3. How well does the method scale to much larger, modern architectures?
>
> A3. Scalability to much larger architectures is primarily limited by computation, not by the certification principle itself. As architectures grow, generating synthesis trajectories and SCP verification scenarios becomes more expensive, yielding the expected performance--cost trade-off under a fixed compute budget; this is consistent with the settings in Tables 3 and 4.
> A key practical contribution of the present framework is the surrogate-based synthesis procedure; see Remark 3.1 and Appendix C. The surrogate substantially reduces synthesis cost by replacing repeated expensive rollouts of the true dynamics during certificate construction, while final PAC verification is still performed on fresh samples from the true dynamics $f$. More broadly, high runtime on larger and more complex architectures remains an open challenge across robustness-certification methods. In this context, the present work contributes both a stochastic formulation and a concrete computational mechanism that makes larger settings substantially more manageable in practice.
> This is also important relative to methods that provide point-wise certificates, which constitute a large portion of the certification literature. Such methods must repeat certification for each individual test sample, so their end-to-end computational cost scales with the size of the test set. By contrast, the present framework produces a single model-level certificate for the whole setting, and surrogate-based synthesis further reduces the computational burden. We will make this scalability perspective more explicit in the revised manuscript.

---

> > ### Author Rebuttal · Reviewer_Nsqz · 2026-04-01
> >
> > I thank the authors for addressing the raised questions.

---

> > > ### Author Response · Authors · 2026-04-04
> > >
> > > We sincerely appreciate your helpful feedback on our paper and follow-up rebuttal. We would be happy to provide any additional clarification that may be useful. If our responses have satisfactorily resolved the concerns raised in your review, we would be very grateful if you would consider updating the final score accordingly.

---

### Official Review · Reviewer_Desb · 2026-03-09

**Soundness:** 2
**Presentation:** 3
**Significance:** 2
**Originality:** 2
**Overall Recommendation:** 4
**Confidence:** 2

**Summary:**

This paper reformulates the problem of certifying whether a model can maintain accuracy under poisoning or perturbations during stochastic training as a safety verification problem for stochastic dynamical systems. The training process is modeled as a discrete-time stochastic system, and barrier certificates are used to separate safe and unsafe parameter regions. These certificates are synthesized with neural networks, and PAC-style probabilistic guarantees are obtained via scenario-based optimization. The overall goal is to compute a certified radius under which the model is $(\alpha, \rho)$-certified accurate with confidence level $1-\beta$.

**Compliance With Llm Reviewing Policy:**

Affirmed.

**Final Justification:**

The author has provided answers to my questions, and I will consider improving my score in the future

**Key Questions For Authors:**

See weaknesss

**Limitations:**

Yes

**Strengths And Weaknesses:**

Strengths:

1:This work attempt to provide formal probabilistic certificates for both training-time and test-time perturbations, whichi is an important and still relatively underexplored part of the robustness verification literature.

2:While the main building blocks of the approach are not individually new, their systematic combination into a complete modeling-learning-verification-PAC pipeline for robustness certification appears to be novel.

Weakness:

1.The probability in CCP/SCP is defined with respect to a joint distribution over $(x, \Delta)$. Could the authors clarify how this distribution is intended to be specified in practice? In particular, what is the semantics of sampling $\Delta$: uniform sampling over the  perturbation set, empirical sampling, or sampling induced by a particular attack model?

2.The concentration argument used in Lemma 4.1 appears pointwise in $x$. Could the authors clarify how this is lifted to the feasibility statement required for the full RCP, which is quantified over all relevant $x$ and $t$?

3.In Lemma 4.2, the definition of $V(\eta)$ seems to require clarification. As written, $V(\eta)$ is defined using the condition $\bar{q}_k \leq 0$, which appears to correspond to constraint satisfaction rather than constraint violation. Is this a typographical issue, or am I misunderstanding the intended definition? In addition, it would be helpful to clarify how $t$ is handled in the underlying probability space.

4.The "attack-agnostic" terminology currently appears broader than what is established in the paper. The framework is general with respect to model architectures and training procedures, but the attack model still assumes input-space, label-preserving, fixed-ratio, and $\ell_p$-bounded perturbations. Since label poisoning and mixed feature-label corruption are explicitly left out. It would be better if the authors to narrow this claim or state the scope more precisely.

5.The certification target is the final accuracy evaluated on a fixed held-out test set, rather than a direct population-level generalization guarantee. Stating this limitation explicitly would help readers interpret the scope of the result more precisely.

6.Although the paper discusses computational considerations and introduces approximations to improve tractability, the empirical section does not yet fully characterize scalability with respect to larger models, longer training horizons, or more complex datasets. This would be an important direction for future validation.

---

> ### Author Rebuttal · Authors · 2026-03-31
>
> We are grateful for the reviewer’s feedback and provide our responses below.
>
> Q1. How is this distribution specified in practice and what is the semantics of sampling $\Delta$?
>
> A1. The perturbation distribution plays different roles in synthesis and verification. In synthesis, perturbations are generated by the chosen attack procedure: we construct adversarial perturbations $\Delta_i$ and use the resulting attacked trajectories to train the candidate NNBC, exposing it to hard or near-worst-case behaviors. In verification, however, the PAC/scenario argument does not require a specific attack algorithm or law such as uniform or Gaussian. It only requires the verification pairs $(x_i,\Delta_i)$ to be sampled i.i.d. from a fixed distribution on $X \times \bar{\Omega}$, where $\bar{\Omega}$ is the admissible perturbation set with $\\|\\Delta\\|_p \le \delta$. Thus, one may sample $x_i$ i.i.d. from a fixed law on $X$ and $\Delta_i$ i.i.d. from a fixed law on $\bar{\Omega}$; the PAC guarantee is then valid. Attack-based sampling may also be used if the resulting pairs remain i.i.d.
>
> Q2. The concentration argument in Lemma 4.1 appears pointwise in $x$. How is it lifted to the feasibility statement for the full RCP over all relevant $x$ and $\Delta$?
>
> A2. The issue was a notational omission: the perturbation variable $\Delta$ was suppressed in several places. In the revised manuscript, the verification constraints are written as $q_k(x,\Delta,t)$, and the expectation term as
> $E(x,\Delta,t):=\\mathbb{E}[\\mathcal{B}_\\varphi^*(f(x,\Delta,w),t)\mid x,\Delta]$.
>
> Accordingly, both RCPs are posed over the joint space $(x,\Delta)$ rather than over $x$ alone. Lemma 4.1 assumes a single variance bound $V_m>0$ for all $(x,\Delta)\in X\times\bar{\Omega}$ and all $t\in\mathbb{N}_{1:T}$; hence the bound is uniform over the full verification domain, not pointwise in one fixed $x$. This lifts the concentration argument from fixed $(x,\Delta,t)$ to the full RCP domain. For each fixed $(x,\Delta,t)$, the proof compares the exact conditional expectation with its empirical approximation, with the Chebyshev probability taken only over the auxiliary i.i.d. samples $w_1,\dots,w_M$. Lemma 4.1 then implies that feasibility of RCP$_M$ yields feasibility of RCP with confidence at least $1-\beta_0$.
>
> Q3. In Lemma 4.2, it appears to correspond to constraint satisfaction rather than violation.
>
> A3. This is a typo in the definition of the violation quantity in Lemma 4.2. The intended event is constraint violation; thus, (52) is $\bar q_k(x,\Delta,t) > 0$.
>
> Q4. The “attack-agnostic” appears broader than what is established, since the framework still assumes the threat model in Definition 2.1.
>
> A4. We agree that “attack-agnostic” should be understood relative to the threat model in Definition 2.1. Our claim is that, within this threat model, the framework is agnostic to the specific attack-generation algorithm used to produce admissible perturbations. The certification pipeline is formulated in terms of admissible perturbation sets and attacked samples, rather than a particular attack construction. This contrasts with prior work, where certificates are often tied either to a specific attack mechanism or to a single attack stage. By contrast, our framework handles both train-time poisoning and test-time evasion within one verification framework, without being tied to a particular attack-generation strategy inside the adopted $\ell_p$-bounded setting.
>
> Q5. The certification target is on a fixed held-out test set, rather than a direct population-level generalization guarantee.
>
> A5. The reviewer is correct: the certificate is defined through the empirical functional $\mathcal{G}$ on a fixed held-out evaluation set, not as a population-level guarantee. We certify the robustness of a stochastic training-and-evaluation pipeline under a given perturbation budget with respect to that evaluation set, so the certified radius is evaluation-set-specific. A population-level guarantee would require an additional generalization layer linking the held-out functional to the data-generating distribution; beyond training stochasticity, one would also need to control evaluation-set sampling variability. That is an important but different problem from the one addressed here. We agree this is a valuable future direction, but it is outside the scope of the present paper.
>
> Q6. The empirical section does not yet fully characterize scalability.
>
> A6. We agree that the scalability aspect should be stated more explicitly. The empirical study already includes datasets of increasing complexity, multiple model classes, and both train-time and test-time certification settings, but these results are not yet organized clearly from a scalability perspective. In the revised manuscript, we will make this aspect more explicit in the main text and highlight more clearly the corresponding discussion on runtime, sampling, and surrogate-based synthesis.

---

> > ### Author Rebuttal · Reviewer_Desb · 2026-04-02
> >
> > The author has provided answers to my questions, and I will consider improving my score.

---

> > > ### Author Response · Authors · 2026-04-03
> > >
> > > We sincerely appreciate your careful reading of our rebuttal and your reconsideration of the score. Your comments have been very valuable in helping us improve the clarity and overall presentation of the paper.

---

### Official Review · Reviewer_nv2Q · 2026-03-12

**Soundness:** 2
**Presentation:** 2
**Significance:** 3
**Originality:** 3
**Overall Recommendation:** 4
**Confidence:** 3

**Summary:**

This paper studies the problem of ensuring safety in neural network training under adversarial attacks. The proposed method learns a neural barrier certificate from stochastic training trajectories and then formally validates it with scenario optimization to certify the largest perturbation radius under which the final model remains accurate with specified confidence and violation probability.

**Compliance With Llm Reviewing Policy:**

Affirmed.

**Final Justification:**

My concerns are addressed. Therefore the score is raised.

**Key Questions For Authors:**

It would be helpful for the authors to explicitly specify what distributions of what random variables are considered in the key probabilistic quantities. This is particularly important for the proof of Lemma 4.1, around (42).

**Limitations:**

Yes

**Strengths And Weaknesses:**

Strengths:

1. The problem studied in the paper is very important.
2. The authors show promising performance in the experiment compared to existing methods.

Weaknesses:

1. Presentation quality is questionable. There are many places with typos, some serious enough to cause confusions. For example, regarding constraint $q_4$, $\epsilon_M$ appears in (15) and (19), but does not appear in (9) and (22). It is also confusing that, since $b_3$ and $\epsilon_M$ are both learned parameters, why they cannot be combined.
2. On top of that, there are many basic definition and math issues. For example, in (24), the probability is taken over the joint distribution of $(x,\Delta)$, so it does not make sense to say it satisfy a relation for all $x$.

---

> ### Author Rebuttal · Authors · 2026-03-31
>
> We are grateful for the reviewer’s feedback and provide our responses below.
>
> W1. Regarding constraint $q_4$, $\epsilon_M$ appears in (15) and (19), but does not appear in (9) and (22). It is also confusing that, since $b_3$ and $\epsilon_M$ are both learned parameters, why they cannot be combined.
>
> A1. We thank the reviewer for raising this point. The issue is not a typo in the equations themselves, but that the role of $\epsilon_M$ was not made sufficiently explicit when moving from the exact formulation to its data-driven approximation. In particular, the third BC condition in Definition 2.6, namely (9), is stated in terms of the exact conditional expectation, so no finite-sample approximation term appears there. By contrast, $\epsilon_M$ is introduced only after replacing that exact expectation by an empirical average over $M$ samples; this is why it appears in the NNBC training loss (15), in the verification constraint $q_4$ in (19), and in the relaxed problem (22), all of which belong to the data-driven synthesis/verification layer rather than the exact BC formulation.
> For the same reason, although $b_3$ and $\epsilon_M$ could be merged mathematically without affecting validity, we keep them separate for interpretability: $b_3$ belongs to the original BC condition already at the exact level in (9), whereas $\epsilon_M$ arises only from the finite-sample approximation introduced in (15) and inherited in (19) and (22).
>
> W2. In (24), the probability is taken over the joint distribution of $(x,\Delta)$, so it does not make sense to say it satisfy a relation for all $x$.
>
> A2.
> We agree that the original notation in the verification section was imprecise: in the previous version, the perturbation variable $\Delta$ was suppressed in several places for brevity, but once the CCP probability is defined over the joint variable $(x,\Delta)$, the formulation must be stated consistently on the corresponding joint space.
>
> Accordingly, we now introduce the admissible perturbation sets already in Definition 2.1:
> $\\Omega := \\{\\Delta \\in \\mathbb{R}^{r\\times m} \\mid \\|\\Delta\\|_p \\le \\delta\\}$ (resp., $\\Omega'$),
> and use the generic notation $\\bar{\\Omega}\\in\\{\\Omega,\\Omega'\\}$ in the verification stage before (16)–(20). The constraint functions are now written explicitly as $q_k(x,\\Delta,t)$.
>
> Then, the expectation term is written as
> $
> E(x,\\Delta,t):=\\mathbb{E}\[\\mathcal{B}_\\varphi^*(f(x,\\Delta,w),t)\\mid x,\\Delta\]
> $,
> as introduced in (19).
>
> The optimization problems are revised accordingly. The RCP is now posed over the joint space $X\\times\\bar{\\Omega}$, with constraints of the form
> $
> q_k(x,\\Delta,t)\\le 0$, $\\forall (x,\\Delta)\\in X\\times\\bar{\\Omega}$,  $\\forall t\\in\\mathbb{N}_{0:T},
> $
> as stated in (21). For the CCP, we remove the misleading pointwise interpretation over $x$ and state the probability only with respect to the joint distribution of $(x,\\Delta)$, exactly as in (24). Finally, in the SCP, we sample joint scenarios $(x_i,\\Delta_i)$ and rewrite the sampled sets as
> $\\mathcal{I}' \\subseteq X_0\\times\\bar{\\Omega},$
> $\\mathcal{U}' \\subseteq X_u^\\alpha\\times\\bar{\\Omega},$
> $\\mathcal{X}' \\subseteq X\\times\\bar{\\Omega},$
> so that the scenario constraints are imposed on sampled pairs $(x_i,\\Delta_i)$ rather than on states alone, as reflected in (25).
>
> Q1. What distributions of what random variables are considered in the key probabilistic quantities? This is particularly important for the proof of Lemma 4.1, around (42).
>
> A3.
> We thank the reviewer for this helpful comment. We agree that the paper involves several distinct probabilistic statements, and that the underlying source of randomness should be stated explicitly in each case.
> Accordingly, in the revised manuscript we will clarify the main probability statements as follows. First, after Definition 2.2, we will state that the probability in
> $\mathbb{P}\big[\mathcal{G}(x(T)) \ge \alpha\big] \ge \rho$
> is taken over the stochasticity induced by the dt-SDS in (5), namely the randomness entering the training process through the sequence $w(0),\dots,w(T-1)$, which in turn induces the randomness of the terminal state $x(T)$.
> Second, in the chance-constrained formulation (24), we will state explicitly that the probability in
> $\mathbb{P}\left[\bar q_k(x,\Delta,t)\le 0\right]\ge 1-v$
> is taken over the joint distribution of $(x,\Delta)$ on $X\times\bar{\Omega}$. Consistently, in the scenario program (25), the scenarios $(x_i,\Delta_i)$ are sampled i.i.d. from that same joint distribution.
> Third, and most importantly for Lemma 4.1 around (42), we will clarify that the probability in the Chebyshev bound is taken only over the auxiliary i.i.d. samples $w_1,\\dots,w_M$ used to approximate the one-step conditional expectation, for each fixed $(x,\\Delta,t)$, and not over the scenario space $(x,\\Delta)$.
>
>
> We sincerely thank the reviewer for the constructive feedback, which significantly helped improve the manuscript.

---

> > ### Author Rebuttal · Reviewer_nv2Q · 2026-04-04
> >
> > I would appreciate if the authors could explicitly write down the changed equations in their changed forms in A2.

---

> > > ### Author Response · Authors · 2026-04-05
> > >
> > > We thank the reviewer for the careful reading of our rebuttal and for the helpful follow-up comment.
> > > As requested, we explicitly provide the revised equations below in their updated form.
> > >
> > > Since the CCP in (24) is defined over the joint variable $(x,\Delta)$, the verification stage is now formulated consistently over the joint space throughout.
> > >
> > > First, in Definition 2.1, we now explicitly introduce the admissible perturbation sets as
> > >
> > > $$
> > > \Omega := \\{ \Delta \in \mathbb{R}^{r\times m} \mid \\|\Delta\\|_p \le \delta \\},
> > > $$
> > >
> > > $$
> > > \Omega' := \\{ \Delta' \in \mathbb{R}^{r'\times m} \mid \\|\Delta'\\|_p \le \delta' \\},
> > > $$
> > >
> > > and use the generic notation $\bar{\Omega}\in\\{\Omega,\Omega'\\}$ in the verification stage. Accordingly, the constraint functions in (16)-(20) are now written as $q_k(x,\Delta,t)$, where
> > > $$
> > > q_k : \mathbb{R}^d \times \bar{\Omega} \times \mathbb{N}_{0:T} \to \mathbb{R}, \qquad \forall k \in [5],
> > > $$
> > >
> > > and the expectation term in (19) is revised to
> > > $$
> > > E(x,\Delta,t):=\mathbb{E}\left[\mathcal{B}_\varphi^*(f(x,\Delta,w),t)\mid x,\Delta\right].
> > > $$
> > >
> > > The RCP in (21) is correspondingly reformulated over the joint space $X\times\bar{\Omega}$ as
> > >
> > > $$
> > > \text{RCP:}
> > > $$
> > >
> > > $$
> > > \min_{\eta_{k'}} \qquad \eta_1+\eta_3-\eta_2
> > > $$
> > >
> > > $$
> > > \text{s.t.} \qquad q_k(x,\Delta,t) \le 0,
> > > $$
> > >
> > > $$
> > > \forall (x,\Delta) \in X\times\bar{\Omega}, ~ \forall t \in \mathbb{N}_{0:T},
> > > $$
> > >
> > > $$
> > > \\tilde{b} _ {k'} = \\tilde{b} _ {k'} ^ * + \\eta_{k'} , ~ \\tilde{b} _ {k'} \in \mathbb{R}_{>0}.
> > > $$
> > >
> > > $$
> > > \forall k \in [5], ~ \forall k' \in [3].
> > > $$
> > >
> > > To handle the expectation term in (19), we now use the empirical approximation
> > > \\[
> > > \bar{E}(x,\Delta,t):=\frac{1}{M}\sum_{j=1}^M \mathcal{B}_\varphi\left(f(x,\Delta,w_j), t\right) + \epsilon_M^*,
> > > \\]
> > >
> > > which yields the following relaxed $\mathrm{RCP}_M$ problem.
> > > $$
> > > \text{RCP}_M:
> > > $$
> > >
> > > $$
> > > \min_{\eta_{r,k'}} \qquad \eta_{r,1}+\eta_{r,3}-\eta_{r,2}
> > > $$
> > >
> > > $$
> > > \text{s.t.} \qquad \bar{q}_k(x,\Delta,t) \le 0,
> > > $$
> > >
> > > $$
> > > \forall (x,\Delta) \in X\times\bar{\Omega}, ~ \forall t \in \mathbb{N}_{0:T},
> > > $$
> > >
> > > $$
> > > \\tilde{b} _ {k'} = \\tilde{b} _ {k'}^* + \eta_{r,k'} , ~ \\tilde{b} _ {k'} \in \mathbb{R}_{>0},
> > > $$
> > >
> > > $$
> > > \forall k \in [5], ~ \forall k' \in [3],
> > > $$
> > >
> > > $$
> > > \text{where } \bar{q}_ j:=q_ j,~ j\in\\{1,2,3\\},
> > > $$
> > >
> > > $$
> > > \bar{q}_4(x,\Delta,t) := \bar{E} (x, \Delta, t) - \kappa^{*} \mathcal{B} _\varphi(x, t-1) - \\tilde{b} _ 3,
> > > $$
> > >
> > > $$
> > > \bar{q}_5(x,\Delta,t):= \\tilde{b} _ 1 \kappa^{*T} + \\tilde{b} _ 3 T - \\tilde{b} _ 2 ( 1-\frac{\rho}{1-\beta_0} ).
> > > $$
> > >
> > > Consistently, Lemma 4.1 is also revised so that the variance bound is stated over the same joint domain:
> > > $$
> > > \text{Assume there exists a uniform } V_m>0 \text{ such that }
> > > \mathrm{Var}\left[\mathcal{B}\left(f(x,\Delta,w),t\right)\middle|x,\Delta\right]\le V_m
> > > $$
> > >
> > > $$
> > > \text{for all } (x,\Delta)\in X\times\bar{\Omega} \text{ and all } t\in\mathbb{N}_{1:T}.
> > > $$
> > >
> > >
> > >
> > > The CCP in (24) is then written directly with probability taken over the joint distribution of $(x,\Delta)$ as
> > > $$
> > > \text{CCP:}
> > > $$
> > >
> > > $$
> > > \min_{\eta_{c,k'}} \qquad \eta_{c,1}+\eta_{c,3}-\eta_{c,2}
> > > $$
> > >
> > > $$
> > > \text{s.t.} \qquad \mathbb{P} [ (x,\Delta)\in X\times\bar{\Omega}:~\bar{q}_k(x,\Delta,t)\le 0 ]\ge 1-v, ~ \forall t \in \mathbb{N} _ {0:T},
> > > $$
> > >
> > > $$
> > > \\tilde{b} _ {k'} = \tilde{b} _ {k'} ^ * + \eta_{c,k'} , ~ \tilde{b} _ {k'} \in \mathbb{R}_{>0},
> > > $$
> > >
> > > $$
> > > \forall k \in [5], ~ \forall k' \in [3].
> > > $$
> > >
> > > Finally, in the SCP, we now sample joint scenarios $(x_i,\Delta_i)$ and use the joint sampled sets
> > > \\[
> > > \mathcal{I}' \subseteq X_0\times\bar{\Omega},\qquad
> > > \mathcal{U}' \subseteq X_u^\alpha\times\bar{\Omega},\qquad
> > > \mathcal{X}' \subseteq X\times\bar{\Omega},
> > > \\]
> > > so that the SCP in (25) is imposed on sampled pairs $(x_i,\Delta_i)$ as
> > > $$
> > > \text{SCP:}
> > > $$
> > >
> > > $$
> > > \min_{\eta_{s,k'}} \qquad \eta_{s,1}+\eta_{s,3}-\eta_{s,2}
> > > $$
> > >
> > > $$
> > > \text{s.t.} \qquad \bar{q}_k(x_i,\Delta_i,t) \le 0,
> > > $$
> > >
> > > $$
> > > \forall (x_i,\Delta_i) \in \mathcal{Z} _ {k},~ \forall t \in \mathbb{N}_{0:T},
> > > $$
> > >
> > > \\[
> > > \mathcal{Z}_1 = \mathcal{Z}_4 = \mathcal{X}', ~ \mathcal{Z}_2 = \mathcal{I}', ~ \mathcal{Z}_3 = \mathcal{U}',
> > > \\]
> > >
> > > $$
> > > \\tilde{b} _ {k'} = \tilde{b} _ {k'} ^ * + \eta_{s,k'} , ~ \tilde{b} _ {k'} \in \mathbb{R}_{>0},
> > > $$
> > >
> > > $$
> > > \forall k \in [5],~ \forall k' \in [3].
> > > $$
> > >
> > > We have revised the manuscript accordingly so that the verification stage is now consistent with the joint formulation above. We sincerely thank the reviewer for pointing out this issue. It stemmed from an imprecise simplification of notation, which in turn led to inconsistencies in the verification formulation. This comment was very helpful in allowing us to correct the presentation and state the verification stage more precisely throughout.

---

### Decision · Program_Chairs · 2026-04-30

**Decision:**

Accept (regular)

**Comment:**

This paper finally received three Weak accepts and one Weak reject. After the rebuttal, Reviewers Desb and Nsqz marked their concerns as fully resolved; Reviewer Desb invited a corresponding score update, while Reviewer Nsqz likewise acknowledged resolution but noted that certain limitations tempered a score increase. Reviewer 2NVM remained partially resolved and maintained a Weak accept. Reviewer nv2Q remained partially resolved and maintained a Weak reject, requesting that revised equations and joint-space verification notation be incorporated explicitly into the manuscript.

Considering the overall supportive consensus among reviewers and the substantive rebuttal, including consistent joint ((x,\delta)) formulations and clarified probability spaces, Lemma-level fixes and explicit randomness assumptions, a narrowed and precise scope for “attack-agnostic” claims. The AC recommends accepting this paper. Congratulations!

When preparing the camera-ready version, the authors should fold rebuttal clarifications into the main paper. They should align claims tightly with the adopted threat model and semantics of sampling in CCP/SCP, and expand limitations regarding evaluation-set conditioning, computational scalability, and verification failure or inconclusive outcomes, as discussed in the reviews and rebuttals.